# Enabling high throughput deep reinforcement learning with first principles to investigate catalytic reaction mechanisms

**Tian Lan[1], Huan Wang[1] & Qi An [ID] [2] ✉**

Exploring catalytic reaction mechanisms is crucial for understanding chemical processes, optimizing reaction conditions, and developing more effective catalysts. We present a reaction-agnostic framework based on high-throughput deep reinforcement learning with first principles (HDRL-FP) that offers excellent generalizability for investigating catalytic reactions. HDRL-FP introduces a generalizable reinforcement learning representation of catalytic reactions constructed solely from atomic positions, which are subsequently mapped to first-principles-derived potential energy landscapes. By leveraging thousands of simultaneous simulations on a single GPU, HDRL-FP enables rapid convergence to the optimal reaction path at a low cost. Its effectiveness is demonstrated through the studies of hydrogen and nitrogen migration in Haber-Bosch ammonia synthesis on the Fe(111) surface. Our findings reveal that the Langmuir-Hinshelwood mechanism shares the same transition state as the Eley-Rideal mechanism for H migration to $NH_2$, forming ammonia. Furthermore, the reaction path identified herein exhibits a lower energy barrier compared to that through nudged elastic band calculation.

Understanding the reaction path in catalytic reactions is of paramount importance for enhancing our comprehension of chemical processes, refining reaction conditions, and developing robust catalysts[1–3]. Insights into the reaction mechanism afforded by the reaction path offer a significant avenue for creating more selective and efficient catalysts[4]. Such information is critical for the optimization of catalyst design, reduction of side reactions, and enhancement of selectivity through activation energy control[5]. Moreover, details about optimal reaction conditions, such as temperature, pressure, and reactant concentration, can be deduced from the reaction pathways, thereby boosting yield and efficiency[6]. However, many challenges hinder the exploration of reaction pathways in catalytic reactions, including the complexity of multistep reactions and short-lived intermediates, the dynamic and heterogeneous nature of many catalytic reactions[7,8], and the intricacies of accurately studying reaction pathways experimentally[9].

Artificial Intelligence (AI) has recently emerged as a pivotal tool in various research fields[10–12], including chemistry[13,14]. Although several machine learning algorithms have been applied to chemical reactivity, most remain mechanism-agnostic, requiring a human-interpretable connection between features and the phenomenon of interest. A broad approach towards elucidating unknown mechanisms might be enumerating all possible reaction pathways from reactants to products, but the vast combinatorial possibilities make this impractical, leading to a 'combinatorial explosion'. The application of reinforcement learning (RL) holds substantial potential for enhancing the exploration of reaction networks and mechanistic investigations[15,16]. RL involves an agent tasked with identifying plausible reaction pathways through interactions with a defined environment over time. Instead of laboriously screening all potential reaction steps, RL can navigate reaction networks in an automated manner[15,16].

Although deep reinforcement learning (RL) is considered as one ultimate epitome of exploring unknown reaction mechanisms in an automated and first-principles way, there are many scientific and engineering challenges to the use of RL for this difficult task. For

[1]Salesforce A.I. Research, Palo Alto, CA, USA. [2]Department of Materials Science and Engineering, Iowa State University, Ames, IA, USA. ✉e-mail: qan@iastate.edu

example, in RL, the sequence of data observed by the learning agent could be non-stationary and strongly correlated[17]. These factors can significantly complicate the learning process. In addition, the finite-horizon rollout in RL may introduce bias when estimating the value function[18]. The aggregation of data into an experience replay memory may help to reduce non-stationarity, but it necessitates extensive computational resources and memory, and restricts the methods to off-policy algorithms[17]. Those challenges are further exacerbated when dealing with the unknown complex chemical reaction mechanisms, where the potential energy landscape (PEL) of chemical systems is characterized by strong nonconvexity, high noise, and high dimensionality. These characteristics present a significant hurdle for RL optimization. For example, RL exploration usually meets a great number of different reaction pathways, leading to a complex energy landscape with numerous local minima and similar energy barriers. These features can trap the policy exploration, preventing it from effectively escaping and causing the convergence to either be impossible or excessively slow.

The great complexity of investigating catalytic reaction mechanisms poses a substantial challenge for the generalizable usage of RL across a variety of catalytic reactions. Therefore, much of the RL research on chemical reactions has instead primarily focused on modeling the environment with a semi-empirical representation based on specific chemical reactions. These methods heavily rely on the tedious design of special state vector encodings, heuristic action rules, or nonlinear transforming reward functions[13,15,16], which can severely limit the scope and potential applications of the RL algorithm. Particularly, it does not offer a transferable solution since it tends to be applicable to a specific reaction and requires a considerable amount of empirical design, hyperparameters tuning, and semi-empirical calculations. Moreover, the simplified RL representation of environment usually requires restricting the exploration to a predefined set of reaction networks, thereby hindering the discovery of unknown elementary reaction mechanisms. For instance, our prior RL study[15] developed an encoded state vector consisting of 23 elements, accommodating a total of 20 surface sites within the catalytic surface. The value of each entry represents various absorbents and are encoded by integers. In addition, the actions connecting states are defined in terms of some specific chemical reactions in which the atomic motion are not explicitly involved in these reactions. This type of RL application heavily depends on the domain knowledge and lacks generality when using it in other complex reactions. Therefore, it is limited to managing intricate reaction paths defined by a predetermined assortment of diverse reaction mechanisms. Meanwhile, the elementary task of probing undiscovered reaction mechanisms remains elusive.

To tackle these issues, we have developed a reaction-agnostic, high-throughput, and fast-converging RL framework, termed high-throughput deep reinforcement learning with first principles (HDRL-FP), to autonomously explore catalytic reactions paths and mechanisms. The reaction-agnostic nature of HDRL-FP arises from its independence from the need for human experts to design specific RL representation of environment (e.g., states, actions, or rewards) for a particular reaction. Instead, the RL environment is solely built on atomic positions, which are then mapped to the potential energy landscape (PEL) derived from first principles. The excellent generalizability and cost-efficiency of our framework are primarily a result of the high-throughput capacity enabled by the pioneering architecture of HDRL-FP. This framework is fundamentally different from the recent development of other parallel RL architectures[17–21], but facilitates the fast running of thousands of concurrent RL simulations on a single graphics processing unit (GPU)[22]. The massive number of parallel agents operated by HDRL-FP diversifies the exploration of environment into numerous uncorrelated regions, resulting in significant improvement of the training stability and a dramatic

reduction of runtime in our generalizable RL environment for chemical reactions.

Then, we adopt HDRL-FP to predict a reaction path for the vital hydrogenation step in the Haber-Bosch (H-B) process on the Fe(111) surface. We also examine $N/N_2$ diffusion related reaction steps to demonstrate the generalizability of the HDRL-FP framework. This H-B process plays a critical role in Earth's nitrogen cycle and accounts for over 2% of global energy usage, producing an annual yield of 160 million tons of ammonia[23–27]. Despite a century of intensive research aimed at enhancing the performance of the H-B process, advancements have been slow. Our developed HDRL-FP framework has the potential to contribute significantly to the optimization of this process, potentially reducing production costs and $CO_2$ emission, and facilitating the establishment of smaller and more widespread plants. Therefore, the framework highlights its effectiveness and potential for predicting complex chemical reaction pathways.

## Results and discussion
### Framework development
Central to the deep RL model is a policy of selecting an action $a_t$ given a state $s_t$ at time t, denoted as $\pi_{\theta_p}(a_t|s_t)$, which is represented by a deep neural network with parameters $\theta_p$. We model the evolution of the chemical reaction path as a Markov decision process (MDP)[28], defined by a state space $S$, an action space $A$, a probability transition $P(s_{t+1}|a_t,s_t)$, and a reward function $r$. The reward function, $r$, is typically defined in the combined space $S \times A$. Unlike other methods, HDRL-FP does not require any reaction-specific encoding of atomic or spatial information to define the environment states and actions[15]. Instead, we simply define the states by using the Cartesian coordinates of atom positions. Consequently, the states are represented by the normalized coordinates of the migrating atom and the its Euclidean distance to the target position in the final product, i.e., $s_t = (x_t/L_x, y_t/L_y, z_t/L_z, \text{distance}\{(x_t, y_t, z_t), (x_f, y_f, z_f)\}/D)$ in an orthogonal supercell, where $D = \text{distance}\{(x_0, y_0, z_0), (x_f, y_f, z_f)\}$ is the Euclidean distance of the migrating atom between the reactant (subscript 0) and the final product (subscript f). For multiple migrating atoms, the states will be defined as the concatenated coordinates of these atoms normalized individually.

Actions, which establish connections between the $s_0$ and the $s_f$, are defined as the stepwise movement of the migrating atom in six possible directions within the 3-dimensional grid, i.e., forward, backward, up, down, left, and right. Our framework is adept at handling catalytic reactions involving multiple atoms. To address scenarios with various atom actors, we have devised a two-dimensional action space consisting of 'atom choice' and 'move direction'. For example, the action (1, down) means that the first atom actor moves downward, while (2, left) indicates that the second atom actor moves left. This scalable strategy effectively reduces the learning burden by limiting the total number of actions to the sum of the atom actors and six directions, rather than multiplying the two. Our policy model is equipped with two output heads of actions that share the internal neural network layers: one determines which atom to move, and the other decides the movement direction. In single-atom catalytic reactions, the 'atom choice' aspect is fixed and set to 1. We apply the periodic boundary condition, so when an atom crosses one edge of the simulation box, it re-enters from the opposite edge. This avoids the issue of an atom going missing as it moves across the boundary.

The crucial link between first principles and deep RL lies in the reward system, which is associated with actions and derived from density functional theory (DFT) calculations. We assign a negative reward, $r$, to each action, which depends on the states before and after the action. Specifically, $r = -\Delta E/E_0$, where $\Delta E$ (in eV) is the electronic energy difference between the current state and the reference state, and $E_0$ is a linear rescaling factor used to normalize the reward within a predetermined range. We also define a penalty reward, $r = -1$, as the

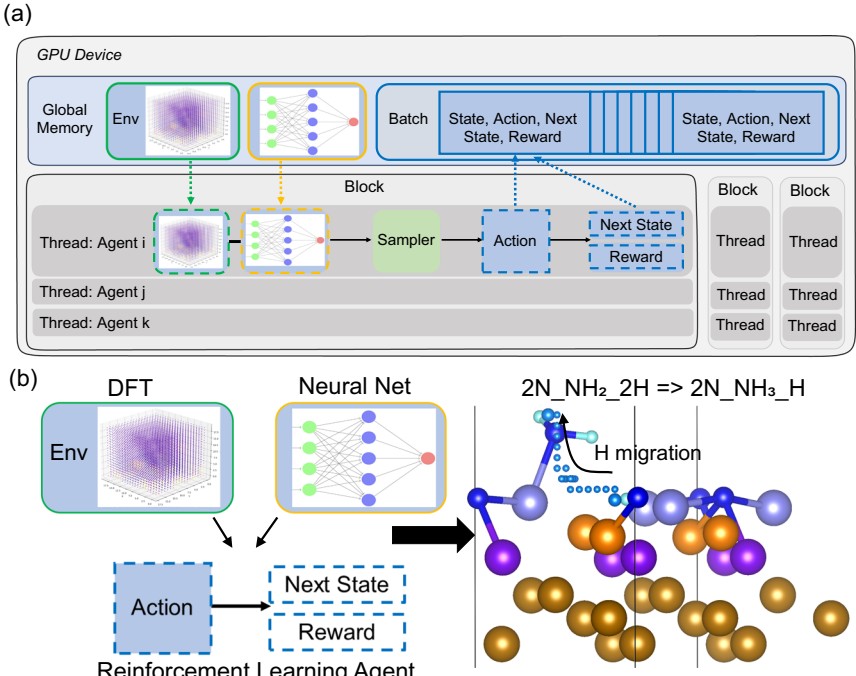

**Fig. 1 | High-throughput deep reinforcement learning (RL) framework for studying hydrogenation reactions in ammonia synthesis on the Fe(111) surface.** **a** A flow chart depicting the first-principles-instructed, high-throughput deep reinforcement learning framework used in the study of the hydrogenation reaction in ammonia synthesis on the Fe(111) surface. Within this framework, computations are organized into graphics processing unit (GPU) blocks. Each block consists of multiple threads to run a concurrent environment, where the states represent the current atomic positions. Each thread operates an atom actor that samples actions and computes rewards. These blocks can access the global GPU memory, which shares the potential energy landscape and deep policy models, as well as stores rollout data for training purposes. Dashed boxes represent the references (not the copies) of the model objects or data placeholders hosted in the global memory shown in the solid boxes. **b** The target scientific questions to be addressed by using the framework in **a** the hydrogenation reaction of ammonia synthesis on Fe(111) surface. The density functional theory (DFT) derived environment is incorporated into the RL agent trained by neutral network to predict the catalytic reaction path and mechanism. The N and H atoms are represented by blue and cyan spheres, respectively. The small cyan spheres represent the trajectory of H migration predicted by framework in **a**.

atom moves very close to other atoms. This definition of the environment serves as a universal descriptor for any chemical reactions and completely decouples the state, action, and reward spaces from any specific reaction. Details about defining the environment (actions, states, and rewards) are in the SI and Table S1.

HDRL-FP executes the entire RL workflow end-to-end on a single GPU, using a single store of data for simulation rollouts, action inference, and training. By leveraging the parallelization capabilities of GPUs, this framework enables cost-efficient, high-throughput computation, running thousands of concurrent RL simulations in parallel on a GPU, and training on large batches of experience. As depicted in Fig. 1a, within the GPU, we execute many replicas of the environment instances in parallel. Each environment instance operates independently on a separate GPU block, maintaining its own reference of the PEL, as computed by DFT, and deep policy models shared by the global memory. Given the large number of blocks typically available in a modern GPU, this framework can execute thousands of environments instances in parallel on a single GPU. Employing this framework, we aim to address the generalized and complex chemical reactions, as shown in Fig. 1b, in which the atomic motions are involved explicitly. In this reaction, we will explore the hydrogen migration mechanism in the H-B ammonia synthesis on the Fe(111) surface. The environment of this specific problem depends entirely on the coordination of the atoms within the system. In each environment instance managed by HDRL-FP, the atom actor can sample the action, transition to the next position (new state), and collect the corresponding reward to navigate through the reaction environment. The deep policy models compute the associated probabilities for the subsequent actions given the new state. Each individual tuple (state, action, and reward) is referred to as

an experience step. The exploration data, consisting of experience steps, is stored in the experience batch in-place in the global GPU memory, avoiding extra data transfer. Additionally, each environment instance supports multiple concurrent threads and allows multiple atom actors to interact with the same environment. Given the concurrent operation of multiple environments, individual environments may reach their terminal states at varying time steps. HDRL-FP automatically resets environments that are finished without causing unnecessary interruption to others still in operation.

Once all environments have provided rollout data of experience steps and the training batch is full, training can be conducted on large batches of experience in-place. Prior to updating the deep policy models, the task coordinator in the framework synchronizes atom actors. In the next iteration, all actors start from the updated policy. This synchronized gradient update approach ensures cohesive training, potentially resulting in a faster convergence.

## Evaluation of agent

We apply HDRL-FP to construct the reaction path for the critical hydrogenation step in $NH_3$ synthesis on the Fe(111) surface, which transitions from $2N\_NH_2\_2H$ to $2N\_NH_3\_H$[15,23]. We consider two potential reaction mechanisms: (1) the Langmuir-Hinshelwood (LH) mechanism, in which both reactants are present on the catalytic surface; and (2) the Eley-Rideal (ER) mechanism, where one H is obtained from the gas phase. We have chosen to focus on this specific reaction for two-fold reasons: (1) It represents one of the most significant rate-determined steps in the complex H-B reaction network on the Fe(111) surface[15,23]; (2) determining its reaction path is not trivial, posing a challenge for the nudged elastic band (NEB) method, especially in the

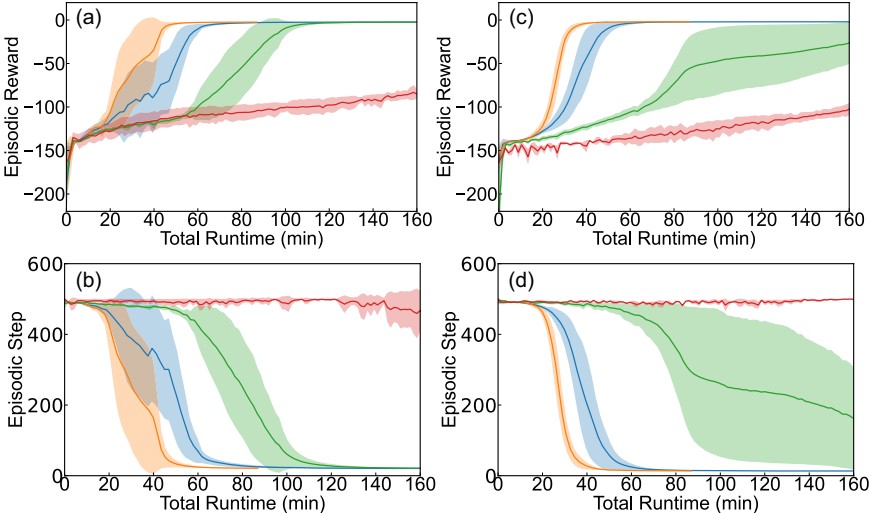

**Fig. 2 | Convergence and learning speed, measured in total runtime using wall-clock minutes, for the high-throughput deep reinforcement learning with first principles (HDRL-FP) method applied to the NH$_2$ to NH$_3$ hydrogenation reaction on Fe(111) surface. a, b** Langmuir-Hinshelwood and **c, d** Eley-Rideal hydrogenation reaction of NH$_2$ to NH$_3$. Different numbers of concurrent environment instances were used – $n = 4$ in red, $n = 20$ in green, $n = 100$ in blue, and $n = 500$ in yellow. The episodic reward is the mean accumulated reward that H atom actors collect from the initial to the terminal state of **a** Langmuir-Hinshelwood and **c** Eley-Rideal, whereas the episodic step is the average total steps to reach the terminal state for **b** Langmuir-Hinshelwood and **d** Eley-Rideal. The model was trained on a single Nvidia A100 GPU. For robustness, the depicted results are averaging over five independent runs from scratch with different initialization seeds and the same hyperparameters. The shadow regions represent the error bar (standard deviation) of five independent runs in all four subfigures. Source data are provided as a Source Data file.

context of the ER mechanism. It is necessary to make an assumption about the initial position of the H atom in the NEB calculations for the ER mechanism[23].

Figure 2a, b displays the convergence speed of the HDRL-FP as it processes the LH reaction as a function of the number of environment replicas, running in parallel. All RL simulations were performed on a single Nvidia A100 GPU, hosted on the Google Cloud Platform (https://cloud.google.com/compute/docs/gpus#%20a100-gpus). The data reveal that, under consistent fixed hyperparameters (Table S2 in SI), the simulations operating with an increased number of concurrent environments attain global convergence faster and more stably. Particularly, simulations with 20, 100, and 500 environment replicas in all separate runs reach the same global optimum within 120, 70, and 45 minutes, respectively. In contrast, simulations involving four (or fewer) environment replicas fail to converge (even after an entire day of training) and fail to reach the terminal state. Our generalizable RL environment with the same hyperparameters is directly applicable to the study of ER reaction mechanism. As depicted in Fig. 2c, d, simulations deploying 100 and 500 environments demonstrate robust and consistent convergence. However, simulations with twenty (or fewer) environments do not exhibit satisfactory convergence. The results highlight the critical role of massive parallelism in RL for effectively exploring a broad range of reaction mechanisms through a generalizable RL environment representation built solely upon atomic positions.

HDRL-FP exhibits exceptional performance in enabling high-throughput RL computation with low cost for catalytic reactions. For example, running 500 environment replicas of LH or ER reactions in parallel, our framework achieves an end-to-end training throughput of 0.23 million experience steps per second on a single GPU. For comparison, the training throughput barely surpasses 10,000 experience steps per second when using 20 environments, a level that already exceeds the throughput limit of most RL solutions. It is noteworthy that the throughput can scale even higher, as it is almost linearly correlated with the number of environments in our framework, until it reaches the limit of the GPU memory. These results demonstrate that our deep policy model can successfully explore

complicated catalytic reaction mechanisms with high portability, scalability, and efficiency.

## Reaction mechanisms of key hydrogenation step in H-B on the Fe(111) surface

Prior research identifies three potential rate-determining steps for the H-B process: dinitrogen dissociative adsorption[29], catalyst surface hydrogenation of adsorbed species[30], and ammonia desorption[31]. Recent studies, however, have emphasized N$_2$ adsorption/desorption, NH$_2$ hydrogenation, and NH$_3$ desorption as the probable rate-determining steps on the Fe(111) surface[23]. Particularly, our recent work accentuates NH$_2$ hydrogenation as the pivotal step with the highest reaction barrier[15]. This is followed by ammonia desorption as the second most critical step, and then N$_2$ adsorption/desorption[15]. The current study focuses on NH$_2$ hydrogenation, aligning with spectroscopy studies detecting primarily NH$_2$ on the Fe(111) surface[32], thus justifying the emphasis on the transition from 2N_NH$_2$_2H to 2N_NH$_3$_H.

We illustrate the 2N_NH$_2$_2H state within the full reaction network of NH$_3$ formation from N$_2$ and H$_2$ (Fig. S1 of SI)[15,23] as the reward-defining reference state in RL. For the LH mechanism, two isolated H atoms are located at shallow-deep (SD) and top-shallow (TS) sites on the Fe(111) surface. For the ER mechanism, we position the migrated H atom approximately 4.6 Å from the Fe surface in a vacuum. The most energetically favorable position for NH$_2$ is the TS site on Fe(111). NH$_3$ forms as H migrates to termination states and moves to the top Fe surface site, energetically lower than the TS site. To connect the initial NH$_2$ and final NH$_3$ states in RL studies for both mechanisms, we strategically reposition NH$_2$ to the top site and focus on H migration in RL calculations. Except for migrated H, all atoms remain fixed, allowing us to derive an unrelaxed potential energy landscape. To accurately identify the transition state, we use insights from RL and conduct NEB calculations, with details provided below.

Figure 3a illustrates hydrogen migration paths for both LH and ER mechanisms as identified by HDRL-FP. Figure 3b, c provide additional detail on atomic configurations for the LH and ER mechanisms, respectively. Here, migrated H atom originates either from surface-

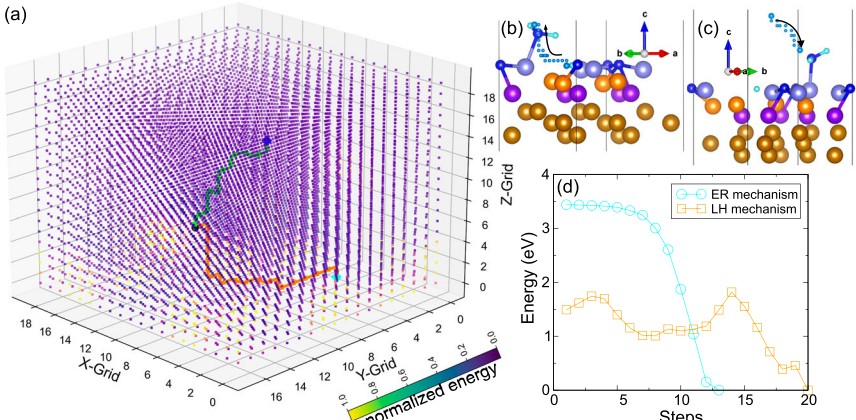

**Fig. 3 | Hydrogen migration pathways and potential energy landscape (PEL) on Fe(111) surface, differentiated by Langmuir-Hinshelwood (LH) and Eley-Rideal (ER) mechanisms. a** Illustration of hydrogen migration pathways with the intricate PEL, originating from 2N_NH₂_2H to 2N_NH₃_H. The path governed by the LH mechanism is designated by the orange trajectory, where the ER mechanism is represented by the light green path. Initial positions for LH and ER are marked by cyan and blue diamonds, respectively, and the terminal position is denoted by a black circle. **b** Detailed visualization of the H trajectory specific to the LH mechanism. **c** Close examination of the hydrogen trajectory as guided by the ER mechanism. **d** Depiction of PEL along the pathways for both LH and ER mechanisms. The Fe atoms on the top, sublayer, deep-layer, and 4th−6th layers are represented by light blue, orange, purple, and bronze spheres, respectively. The N and H atoms are represented by blue and cyan spheres, respectively. The small cyan spheres represent the trajectory of H migration. The color bar indicates the energy levels of the grid points (subfigure **a**) used in the reinforcement learning (RL). The points in subfigure (**a**) are color-coded based on their energy. Source data for **a**, **d** are provided as a Source Data file. The structure files in **b**, **c** are provided in Supplementary Data 1.

adsorbed H (LH) or gas phase H (ER). The H atom, in both cases, follows minimal energy paths, indicated by the darkened trajectories. In the LH mechanism, high-energy zones along the H migration path are due to co-existing adsorbates such as H, N, and surface Fe atoms. The H atom predominantly migrates laterally until near its terminal state, where it moves vertically toward the NH₂ molecule, resulting in NH₃. In contrast, the ER mechanism exhibits a gradual descent of H to its terminal state. The final steps of ER and LH mechanisms bear close resemblance, with the key distinction being the vertical position of the H atom; it descends toward NH₂ in the ER but remains consistently elevated in the LH. It is worth noting that the HDRL-FP calculations are based on a mesh grid of 0.4 Å in the supercell. The convergence of the griding space has been demonstrated by a more complex calculations with 0.2 Å grid spacing. The details of these calculations are discussed in the SI and Figs. S2 and S3 of SI.

Figure 3d displays the potential energy paths for both LH and ER mechanisms as determined by HDRL-FP. For the LH mechanism, the initial configuration exhibits a higher energy relative to the reference energy, as the NH₂ is relocated to a top site with elevated energy. Notably, the H atom surmounts an energy barrier to arrive at a local minimum along the low energy pathway, prior to reaching the terminated state. This observation suggests the presence of an intermediate, lower-energy state along the hydrogen migration reaction pathway derived from HDRL-FP. We subjected this structure to optimization using DFT and discovered it to be 0.15 eV lower in energy compared to the initial 2N_NH₂_2H state. Even after considering the free energy correction at $T$ = 673 K and $P$ = 20 atm, the new state's free energy is found to be 0.11 eV lower than the original state[23].

Our HDRL-FP calculations led to the discovery of a previously unidentified lower-energy state not found in prior research[23]. As shown in Fig. S4 (SI), this state positions both H atoms at the top-shallow-deep (TSD) sites. The previous study[23] used the NEB method to identify the reaction path, with the H atom migrating over a shorter distance. Due to the NEB method's focus on shorter paths, longer migration mechanisms remained elusive. Starting from this new energy state, the H atom surmounts another barrier (~1.0 eV) before reaching the termination state (Fig. 3d). Unlike the LH mechanism, the ER mechanism exhibits consistent energy reduction during H migration due to the high energy state on the migration path exceeding the reaction barrier.

## NEB calculation using RL insight to determine the transition state

The HDRL-FP approach, which fixes all atoms except the migrated one, may yield approximate transition states and reaction energy barriers. To identify the exact transition state and obtain accurate reaction barriers for both LH and ER mechanisms, we deployed NEB simulations using pathways predicted by RL. By utilizing pathways generated by RL, NEB simulations can more effectively pinpoint transition states. Figures 4 and S5 (SI) reveal identical transition state structures for both mechanisms, each with an energy barrier of ~1.40 eV. At $T$ = 673 K and $P$ = 20 atm, free energy correction reduces the barrier to 1.16 eV, making it lower by 0.24 eV than what our prior study suggested[15]. The initial reactant, derived directly from RL paths, lowers its energy after DFT-NEB relaxation. The observation of shared transition states in LH and ER mechanisms suggests a common key barrier, implying similar activation energy and reaction rates, and revealing a shared transition state. This insight suggests a shared key reaction step under realistic conditions. The LH mechanism, though less enthalpically favored, has a lower free energy than the ER configuration at $T$ = 673 K and $P$ = 20 atm (details in SI), indicating its potential plausibility under experimental conditions.

## Free energy diagram for the reaction network of ammonia synthesis on the Fe(111)

Our reconstructed free energy diagram at $T$ = 673 K and $P$ = 20 atm depicts the entire reaction network for $N_2 + 3H_2 = > 2NH_3$, as shown in Fig. 5. A modification from 2N_NH₂_2H to 2N_NH₃_H differentiates it from the prior full reaction network[15]. The discovery of a new configuration (Figure S4 of SI) reduced the relative free energy of 2N_NH₂_2H configuration from 0.07 eV to −0.04 eV, and the transition state energy dropped by 0.35 eV. Consequently, this hydrogenation step no longer serves as the rate-determining step as earlier believed[15]. Instead, the 3N_NH₂_2H configuration's hydrogenation step with a 1.47 eV free energy barrier at 673 K becomes the rate-determining step. Our prior study overlooked this step due to inability in predicting the complex hydrogenation transition state. HDRL-FP thus presents a more plausible reaction path[15]. We estimated the reaction rate at $T$ = 673 K using transition state theory to be 137.7 s⁻¹, nearly doubling the previous estimate of 58.2 s⁻¹. Our predicted rate surpasses experimental values,

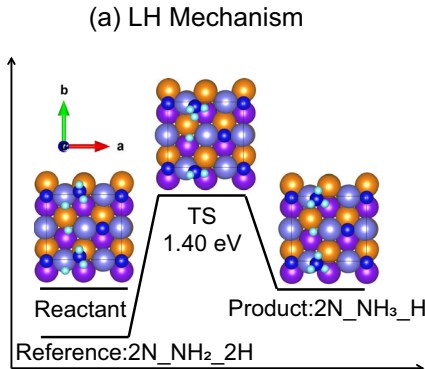

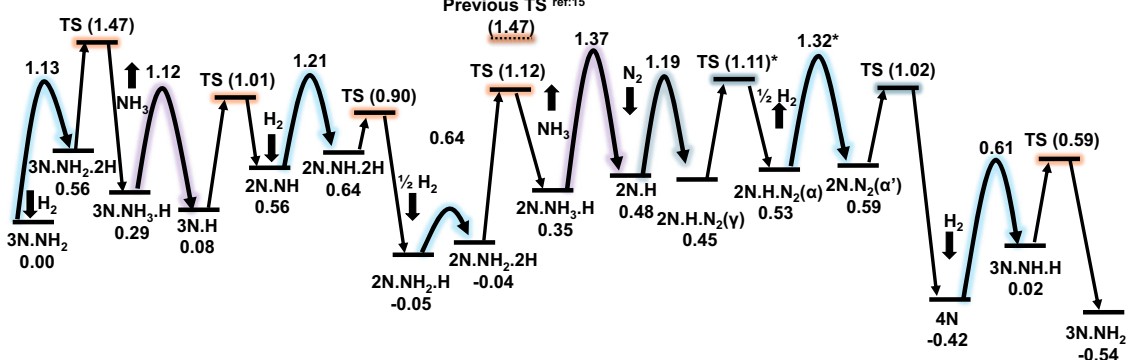

**Fig. 4 | Overview of the transition-state analysis and energy alterations along the reaction pathway of hydrogen migration from 2N_NH₂_2H to 2N_NH₃_H based on reinforcement learning (RL)-guided nudged elastic band (NEB) simulations. a** Langmuir-Hinshelwood (LH) mechanism and (**b**) Eley-Rideal (ER) mechanism. Notably, the view angles differ to better illustrate the two reaction mechanisms. The reactant's reference energy is derived from the new 2N_NH₂_2H configuration identified through the RL. The initial reactants before NEB calculations exhibit a higher energy compared to the reference as they are not optimized in RL. N and H atoms are depicted as blue and cyan spheres, respectively, while Fe atoms in the top, sublayer, deep-layer, and 4th-6th layers are represented by light blue, orange, purple, and bronze spheres, respectively. TS represents the transition state. The structure files are provided in Supplementary Data 1.

**Fig. 5 | Free energy landscape (unit eV) of the reaction network of the Haber −Bosch process on the Fe(111) surface.** The dashed TS from 2N_NH₂_2H to 2N_NH₃_H is from previous study[15], which fails to predict the correct TS and the 2N_NH₂_2H configuration with a lower energy. The thick downward arrows represent the introduction of H₂ into the surface, while the thick upward arrows indicate the desorption of NH₃ from the surface. The purple, light blue, and dark blue shadow lines represent NH₃ desorption from the Fe(111) surface, H addition to the Fe(111) surface, and N₂ addition to the Fe(111) surface, respectively. Part of this figure is from ref. 15.

likely due to the presence of surface impurities that inhibit reaction sites in experiments[26,27]. It is worth noting that the free energy corrections in the present work have been calculated using the harmonic approximation. At elevated temperatures, anharmonic effects are anticipated, though the extent and the temperature at which they become significant vary according to the specific process being examined. Given that our free energy diagram is estimated at a high reaction temperature of 673 K, it becomes essential for future research to explore the impact of anharmonic effects on the free energy barriers. This exploration could be effectively conducted using molecular dynamics techniques[33].

**More case studies**

To enhance the assessment of HDRL-FP's generalizability, we explored two nitrogen diffusion-related processes in the Haber-Bosch (H-B) procedure: (1) the diffusion of nitrogen atoms on the Fe(111) surface, and (2) the diffusion of N₂ molecules. These diffusion processes, involving N and N₂, might correlate with nitrogen coverage on the Fe(111) surface and the adsorption or desorption dynamics of N₂ molecules from it. Understanding the mechanisms and barriers associated with N/N₂ diffusion is crucial. Significantly, the entire study employed a consistent set of hyperparameters (refer to Table S2 in the Supplementary Information), initially established for the hydrogenation analysis discussed in section, evaluation of agent, without any additional adjustments.

Initially, the N atom bridges two apex Fe atoms on the Fe(111) surface. Our findings indicate a diagonal diffusion trajectory, occurring between a top-layer Fe atom and its sublayers as N transitions between bridge sites, as shown in Fig. 6a, b. (This diffusion path aligns with NEB calculations (Fig. S6 of SI) for N diffusion. Both HDRL-FP and NEB predict two energy barriers during N diffusion, with the higher barrier occurring during N's transition through the intermediate region between top and sublayer Fe atoms.

Investigating the performance of our HDRL-FP framework under varying chemical environments is crucial, especially considering the environments in which atoms are allowed to relax. It is important to note that maintaining the configurations of the initial reactants and final products necessitates the implementation of certain constraints. Therefore, we have employed N diffusion as an example to examine this aspect. Figure 6c, d shows a comparative analysis of the unrelaxed and relaxed calculation outcomes. Specifically, Fig. 6c depicts the results from the calculation with nitrogen relaxation along the direction perpendicular to the surface, displaying a diffusion pathway that is strikingly similar to the unrelaxed scenario shown in Fig. 6a. A notable

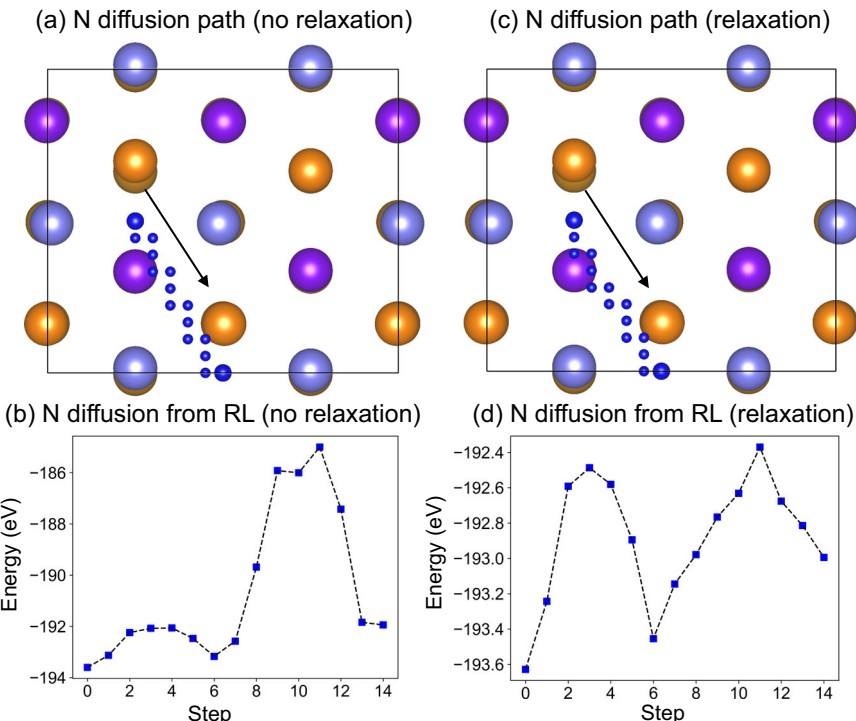

**Fig. 6 | N diffusion pathway on Fe(111) surface and corresponding potential energy profiles determined by reinforcement learning.** The identified N diffusion path on the bare Fe(111) surface as determined by reinforcement learning (RL): (**a**, **c**) The diffusion path for the N atom from one bridge site to the other bridge site. The N atom in the initial and final state is represented by blue sphere, while the small blue spheres represent its positions along the diffusion path as identified by RL. The Fe atoms in the top, sublayer, deep-layer, and 4th–6th layers are represented by light blue, orange, purple, and bronze spheres, respectively. The RL with no relaxation in DFT is displayed in **a** and the RL with relaxation in density functional theory (DFT) is in **c**. **b**, **d** The potential energy profile for the snapshots taken along the diffusion path of N atom. The RL with no relaxation in DFT is shown in **b** and the RL with relaxation in DFT is in **d**. The arrows in **a**, **c** represent the diffusion path of N atom on Fe(111) surface. Source data for **b**, **d** are provided as a Source Data file. The structure files in **a**, **c** are provided in Supplementary Data 1.

observation is that the potential energy landscape in the relaxed calculation (Fig. 6d) exhibits a significantly smoother profile compared to its unrelaxed counterpart (Fig. 6b). Interestingly, both the relaxed and unrelaxed conditions reveal a local minimum along the reaction path, indicative of the nitrogen atom's position between the subsurface and the third Fe layer. This revelation emphasizes the value of unrelaxed calculations in yielding vital insights for chemical reactions, while also offering a more straightforward implementation within the HDRL-FP framework. We also compare the HDRL-FP with other methods of finding reaction path and the details are in the SI.

To evaluate the performance of HDRL-FP for chemical reactions with multiple atoms, we studied $N_2$ molecule diffusion to find the reaction pathway and barriers at top Fe sites. $N_2$ migration for the two-atom scenario involves multiple reaction steps and represents one of the crucial rate-determining steps[23]. Thus, we demonstrate our HDRL-FP model's applicability to two-atom systems through the transition from one γ-$N_2$ configuration (on top Fe atom) to another γ-$N_2$ configuration among the $N_2$ related reactions. Given that the primary focus of the current manuscript is to establish the method and concentrate on reactions related to single-atom motion, we will defer the investigation of other $N_2$-related reaction steps to future research. We conducted extensive DFT simulations considering six degrees of freedom for the two atoms, using a grid spacing of 0.9 Å, notably smaller than the Fe-N bond distance of 1.8 Å. Figure 7 illustrates the varying convergence rates of the HDRL-FP algorithm in processing $N_2$ diffusion, depending on the number of parallel environment replicas. The results demonstrate that simulations with more concurrent environments reach global convergence quicker and more reliably. In particular, simulations with 20, 100, and 500 environment replicas achieve global optimum in 45, 25, and 15 minutes, respectively, whereas those with

four or fewer replicas exhibit convergence challenges. These outcomes again highlight the critical role of extensive parallelism offered by our framework for efficiently exploring diverse reaction mechanisms through a generic RL environment representation based solely on atomic positions. This study also confirms the effectiveness of our approach in handling reactions involving multiple atoms.

Figure 8a displays the RL-predicted path for $N_2$ diffusion from one top Fe site to another. Subsequently, we performed DFT-based NEB calculations to ascertain the reaction barriers and transition states, as depicted in Fig. 8b. The energy barrier for $N_2$ molecular diffusion was found to be 0.52 eV, lower than that for an N atom diffusion on the Fe surface, further evidencing the strength of our computational method in two-atom scenarios.

## Advances in generalizable RL framework for chemical simulations

In this HDRL-FP framework, two significant advances are present: the introduction of a generalizable RL representation for chemical reactions and the development of a high-throughput strategy to support this RL simulation. One of the primary differences between this work and previous RL method[15] lies in defining and modeling the "state" and "action," which are crucial elements in the policy $\pi_{\theta_p}(a_t|s_t)$ that shape the RL environment. Effective environment design, including the specification of states and actions, is fundamental to the success of RL. A well-crafted environment facilitates effective learning, generalization, and transferability of knowledge, ultimately leading to robust and adaptive RL agents capable of solving complex problems. For example, a major progress in the famous AlphaGo work was using raw board position images as states, allowing AlphaGo to capture complex spatial patterns and dependencies inherent in the game[12].

The complexities of investigating catalytic reaction mechanisms present a formidable obstacle to the broad application of RL across various catalytic reactions. Prior research on applying RL to chemical

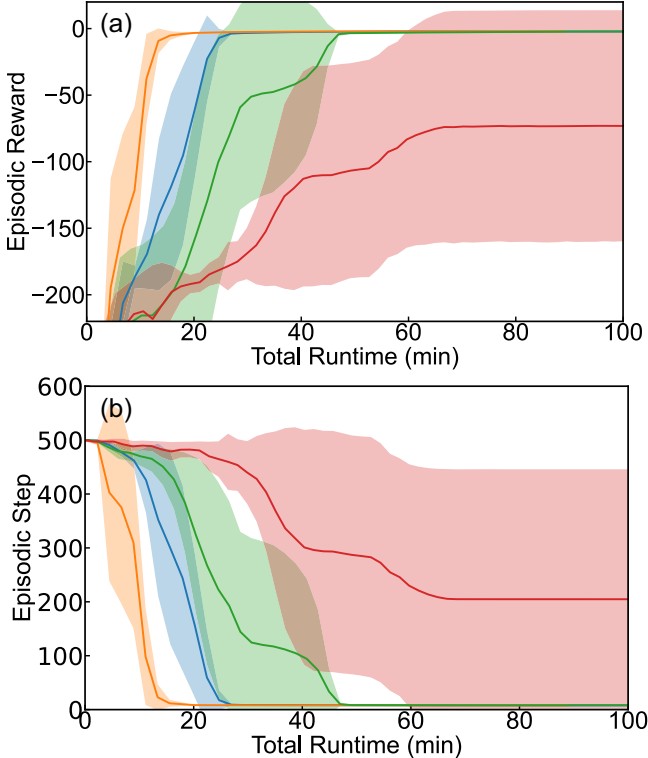

**Fig. 7 | Convergence and learning speed, measured in total runtime using wall-clock minutes, for the high-throughput deep reinforcement learning with first principles (HDRL-FP) method applied to N₂ diffusion on Fe(111) surface.** Different numbers of concurrent environment instances were used – $n = 4$ in red, $n = 20$ in green, $n = 100$ in blue, and $n = 500$ in yellow. Here the episodic reward in panel **a** is the mean accumulated reward that two nitrogen atom actors collect from the initial to the terminal state, and in panel **b** the episodic step is the average total steps to reach the terminal state. The model was trained on a single Nvidia A100 GPU. For robustness, the depicted results are averaging over five independent runs from scratch with different initialization seeds and the same hyperparameters. The shadow regions represent the error bar (standard deviation) of five independent runs in both panel **a** and **b**. Source data are provided as a Source Data file.

reactions has predominantly focused on modeling states and actions through a simplified or semi-empirical representation specific to certain chemical reactions[15,16]. For example, in our previous study[15], we introduced an encoded state vector consisting of 23 elements, accommodating a total of 20 surface sites and three gas species. To address these limitations, our work here introduces a generalizable approach that defines reaction-agnostic states using the Cartesian coordinates of atom positions. States are represented by the normalized coordinates of the migrating atom and its Euclidean distance to the target position. Actions are defined as stepwise movements of the migrating atom in six possible directions within a 3-dimensional grid. This method avoids the laborious and unscalable semi-empirical modeling of reaction-specific environments employed in previous methods.

Our approach, featuring a generalizable representation of states and actions, is boosted by our evolutionary framework, HDRL-FP, for high-throughput (supporting thousands of concurrent RL environments) and low-cost computations (running on a single GPU). This framework significantly diminishes correlations and noise between exploration steps in RL, ensuring training stability, accelerating convergence, and mitigating the need for fine-tuning hyperparameters. Without this efficient computational architecture for high-throughput processing, RL applied to a generalizable environment representation based on atomic coordinates would struggle to converge due to its computational complexity, high noise, strong correlation, and non-stationarity of chemical reactions. Quantitative studies on RL simulations operating with varying numbers of concurrent environments show that simulations with an increased number of concurrent environments achieve global convergence faster and more reliably. This highlights the crucial role of massive parallelism in facilitating effective RL exploration of a wide array of reaction mechanisms through a generalizable environment representation constructed solely from atomic positions.

In summary, we have developed HDRL-FP, an AI framework that can autonomously map complex catalytic reaction paths from scratch. This method circumvents the need for tedious empirical or semi-empirical design of reaction-specific representations of RL environments. Instead, the RL environment is solely reliant on PEL derived from first principles and is intrinsically defined by atomic positions (or configurations). This model's broad applicability and cost-efficiency stem from its capability for high-throughput computations, allowing rapid convergence across a diverse set of catalytic reactions. This feature reduces correlation between RL exploration steps, ensures training stability, accelerates convergence, and alleviates the need for

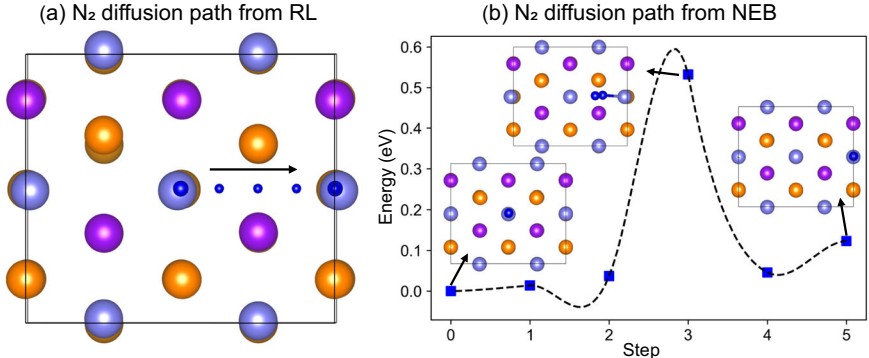

(a) N₂ diffusion path from RL (b) N₂ diffusion path from NEB

**Fig. 8 | Reinforcement learning and nudged elastic band determined N₂ diffusion pathway and reaction barriers on Fe(111) surface.** The identified N₂ diffusion path on the bare Fe(111) surface from the one top site to another top site as determined by reinforcement learning (RL) (**a**), followed by the nudged elastic band (NEB) calculation (**b**) to determine the reaction barriers and transition state. The Fe atoms in the top, sublayer, deep-layer, and 4th–6th layers are represented by light blue, orange, purple, and bronze spheres, respectively. The N atom is represented by the blue sphere. In panel **a**, the black arrow indicates the direction of N₂ diffusion. In panel **b**, the blue squares represent the calculated energetics along the NEB reaction path, while the dashed line represents the corresponding fitting curve. Source data for **b** are provided as a Source Data file. The structure files are provided in Supplementary Data 1.

hyperparameter fine-tuning. HDRL-FP can adapt to a wide range of catalytic reactions depending on the complexity of the environments. Our current approach works well for both non-relaxation environments and relaxation environments with well-defined constraints (e.g., relaxing along one direction). Combining HDRL-FP with proper constraint design will be an important future research direction for exploring catalytic reaction mechanisms in complex environments.

Using $NH_3$ synthesis via hydrogenation reactions on the Fe catalyst as an illustrative example, HDRL-FP accurately predicted a crucial hydrogenation step's reaction path in an hour on a single GPU. Remarkably, this path exhibits a lower reaction barrier than those predicted by direct NEB calculations[15,23], signifying our HDRL-FP framework's potential in predicting complex chemical reaction pathways. The successful application of HDRL-FP to the H-B enables the investigation of complex catalytic chemical reactions automatically, offering a promising approach for future research and discoveries.

## Methods

### Policy and value networks

The policy network, denoted as $\pi_{\theta_p}(a_t|s_t)$, provides the probability distribution of actions $a_t$ given the state $s_t$. Here the subscript $\theta_p$ represents the set of parameters that define the policy. The current state vector $s_t$, as defined in the Framework Development section, represents a feasible configuration along the reaction path. The value network, $V_{\theta_V}(s_t)$, approximates the expected value of the return under the policy $E_\pi[G_t|s_t]$, given the state $s_t$. Here the return $G_t$ at step t up to the terminal step T is defined as the sum of discounted future rewards $r_{t+1}, r_{t+2}, \ldots, r_T$ with the discount factor $\gamma = 0.99$, $G_t = \sum_{\tau=t+1}^{K} \gamma^{\tau-t-1} r_\tau$. It was collected during rollout sampling, as detailed in the Rollout Sampling section below.

Our policy and value networks are composed of fully connected deep neural networks (DNNs)[15]. These DNNs share a common internal stack of hidden layers but have different final output layers. In this study, the two hidden layers consist of 50 neurons each, activated by the ReLU function. The output layer of policy network has 6 neurons and utilizes the Softmax activation function to map to the probability distribution of the 6 possible actions. The value network, outputs a scaler value that estimates the return. The model parameters $\theta_p$ and $\theta_V$ are initialized randomly, with no prior knowledge about the environment or the reaction domain.

The deep policy and value models in our framework are shared among all environment instances and are hosted in the global memory of the GPU device. Each environment instance runs on a separate GPU block and maintains its own reference to the deep models. The policy and value networks are trained using the actor-critic policy gradient algorithm[17,28]. During training, the parameters $\theta_p$ and $\theta_V$ are updated to maximize the expected sum of discounted future rewards, contingent on successfully reaching the final state. In particular, the value network, $V_{\theta_V}(s_t)$, is utilized to update the policy $\pi_{\theta_p}(a_t|s_t)$, with the policy gradient pointing towards improving the policy to favor actions that yield higher expected returns[15]. To improve the value network, the parameters $\theta_V$ are updated to minimize the mean square error between the current value function, $V_{\theta_V}(s_t)$, and the observed return, $G_t$. We employ proximal policy optimization[34] and the Adam optimizer to compute policy gradients[35].

### Rollout sampling

For each instance of the environment, at each subsequent step $t$, we collect rollout data using Monte Carlo (MC) method. Based on the current state $s_t$, the actor selects an action $a_t$ drawn from the distribution generated by the policy network $\pi_{\theta_p}$. By executing the action, the model transits to the next state $s_{t+1}$ and receives the corresponding reward. The rollout continues until reaching either the terminal state or the maximum step of a rollout, $K$. In our experiment, $K$ is set to be 500. If the environment ends, our framework immediately resets it and starts the next rollout if the total number of steps is still less than $K$. Synchronization among all environment instances is not required at this stage. At the end of each rollout, training samples consisting of tuples $(s_t, a_t, G_t)$ are collected. Once rollout data is gathered from all environments and the training batch is full, we perform training on large experience batches of size of $500 \times N$ in-place, where N is the total number of environment instances. The task coordinator synchronizes all atom actors for all environments before updating the deep policy models, ensuring that all actors start the next iteration with the same updated policy.

### Data transfer efficiency

Our workflow begins with a one-time transfer of data from the CPU host to the global memory of the GPU device. This data comprises environment configuration parameters, deep policy models, the potential energy surface computed from first principles, and placeholders for experience batches containing observations, actions, and rewards. Once transferred, no additional data is copied from the host to the device, minimizing CPU-GPU data communication. Throughout the rollout and training periods, data is collected directly into the placeholders without the need for data transfer or copying between the host and the device, or within the device.

### DFT simulations

DFT simulations were conducted utilizing the VASP software (version 5.4) with plane-wave basis sets for the electronic wavefunction[36–39]. A 500-eV energy cutoff for plane wave expansion was consistently applied across all calculations. The Perdew-Burke-Ernzerhof (PBE) exchange and correlation (xc-) functional[40,41] was employed for describing electronic interactions, while Van der Waals interactions were addressed using the D3 corrections[42]. The Methfessel-Paxton scheme was implemented for electron partial occupation, with a 0.2 eV electronic smearing width. Spin-polarization was incorporated in all calculations to accurately predict Fe atom magnetic moments. Energy convergence for the self-consistent field and force convergence for ionic steps were set at $10^{-6}$ eV and $10^{-3}$ eV/Å, respectively. The first Brillouin zone was sampled through the Γ-centered mesh method, using a $4 \times 4 \times 1$ K-point grid in slab calculations. The simulation setup adheres to previous studies[15,23].

The Fe-bcc(111) slab model comprised a $(2 \times 2)$ unit cell in the (111) plane and six layers perpendicular to the plane. Geometry optimization allowed the top three layers to relax while the bottom three remained fixed. A 15 Å vacuum was included to minimize interactions between period images. The model size effects were evaluated by a nine-layer slab model in the previous study[15]. Phonon modes obtained from the harmonic approximation and finite displacement approach[43,44] were used to calculate the free energy at finite temperature, which in turn was used to compute the vibrational entropy as a function of temperature. The free energy of gas phases ($N_2$, $H_2$, and $NH_3$) was sourced from previous work using the same computational parameters (PBE-D3)[15,23]. In gas phase calculations, the zero point energy (ZPE) is derived from the vibrational levels, which are described as simple harmonic oscillators. The specific heat corrections to the enthalpy are calculated from 0 to T. The entropy (S), which includes contributions from vibrational, rotational, and translational modes, is evaluated from these same levels. This computational setup has been shown to accurately describe the H-B reactions on both Fe-bcc(111) and Fe-bcc(211) reconstructed surfaces[15,45].

Transition state searches between initial and final states were initially carried out using the climbing Nudged Elastic Band (cNEB) method[46], followed by refinement via the Dimer approach[47]. Four images were employed in cNEB calculations. Similar to previous studies[15,23], a negative frequency was obtained to confirm the transition state from these calculations. The setup of DFT simulations in this work is similar to our previous work[15,23].

**Reporting summary**

Further information on research design is available in the Nature Portfolio Reporting Summary linked to this article.

## Data availability

All data generated in this study are provided in the Supplementary Information/Source Data file. The structures for DFT simulations are provided in the Supplementary Data file. These structures are visualized using Vesta (veresion 3.5.8)[48] in this paper. Source data are provided with this paper.

## Code availability

The multi-agent reinforcement learning code is explained in ref. 22 and the WarpDrive (version 2.7) is available at https://github.com/salesforce/warp-drive and hosted by PyPI (the official third-party package repository for the Python programming language) at https://pypi.org/project/rl-warp-drive/. The environment code is available at https://github.com/kungfulan/hdrl-fp[49].

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

## Acknowledgements

Q.A. was supported by the start-up grant from Iowa State University and the simulations were performed at ISU High Performance Computing clusters. We also thank the open-source software projects supported by Salesforce A.I. Research.

## Author contributions

Q.A. and T.L. proposed the idea and designed the work. T.L. and Q.A. performed the DFT and RL simulations, as well as wrote the manuscript. T.L. H.W. and Q.A. analyzed the data and discussed the results. All authors commented on the manuscript.

## Competing interests

The authors declare no competing interests.
