## [Peer Review File · Nature Communications]

Enabling High Throughput Deep Reinforcement Learning with First Principles to Investigate Catalytic Reaction MechanismsREVIEWER COMMENTS

Reviewer #1 (Remarks to the Author):

This work describes the ML method application to NH₂_H reaction on Fe(111) surface. After reading the work, while I appreciate the efforts of using ML methods for catalysis, I feel that the reported progress is incremental and the potential impact to catalysis is not obvious. I therefore do not suggest publication of this work in Nature Comm. Apart from the limited insights, the model is also quite simple --- the surface is fixed without considering the realistic high-temperature surface dynamics; the molecule coverage is limited to one coverage, a fully GCMC is desirable. In particular, the authors have published a few works on the same topic and it is such a big surprise that they still did not figure out the correct TS! Even constrained MD simulations should provide the same answer if they carried out the work carefully enough.

Reviewer #3 (Remarks to the Author):

In this paper, the authors have developed and employed a High-Throughput Deep Reinforcement Learning-based strategy to explore mechanisms for NH₂ hydrogenation in the Haber-Bosch (H-B) process on the Fe(111) surface. The essence of their approach lies in utilizing reinforcement learning, where an agent is tasked with identifying the most efficient pathway connecting reactants and products. These pathways are constructed through the selection of sequences of actions, such as translating an atom on a 3D grid, which, in turn, elicits varying 'rewards' derived from density functional theory (DFT) calculations. By optimizing the rewards received, the agent learns to select the most plausible reaction pathways. The RL pathways are then refined by the Nudged Elastic Band (NEB) method, followed by the calculation of harmonic free energies.

In my opinion, this high-quality research paper represents a significant step forward in enhancing the transferability of RL-based algorithms for exploring chemical reaction networks. It demonstrates progress compared to prior publications on the topic (<https://doi.org/10.1021/jacs.1c08794>, <https://doi.org/10.1021/jacs.7b13409>), making such an approach original. Notably, the use of a metric based on the positions of the migrating atom and the Euclidean distance from reference structures (e.g., reactant, product) enables an agnostic exploration of reaction mechanisms, eliminating the need for system-dependent actions in the algorithm.

The above-mentioned technique has allowed the authors to identify a reaction pathway for NH₂ hydrogenation that is kinetically more favored than the ones previously identified by an RL algorithm based on system-dependent actions, demonstrating the efficacy of the algorithm for this specific system.

The employed DFT setup (PBE+GD3) is standard but robust, adequately describing the structural and energetic properties of iron. Furthermore, previous papers by the authors have already assessed and discussed the performance of PBE-D3 in computing the energy barriers associated with the H-B process (<https://doi.org/10.1021/jacs.7b13409>). The authors have also tested for possible size

effects due to the small dimension of the simulation box.

I believe that this work holds great potential for making a significant impact in the field of catalysis by offering an efficient and transferable algorithm for screening reaction mechanisms. However, it's important to note that while the authors claim applicability to complex catalytic reactions, they have yet to provide concrete evidence to support this claim. Specifically, the technique has only been applied to cases involving one-atom diffusion processes.

Another limitation of the study is that the exploration of reaction mechanisms is performed on an unrelaxed Potential Energy Surface (PES). In particular, the mechanism exploration proceeds in the relaxed chemical environment of the reactants. I would like assurance that this does not strongly influence the computed reaction pathway. For instance, a recent paper by Yang et al. (<https://doi.org/10.1038/s41929-023-01006-2>) has shown how the structural and dynamical relaxation of the surface catalyst in response to substrate adsorption could be determinant in rationalizing the catalytic activity of a chemical system.

Addressing the concerns outlined below would support my recommendation for publication in Nature Communications:

1. How can the High-Throughput Deep Reinforcement Learning with first principles (HDRL-FP) ensure that the selected pathway is indeed the optimal one, considering the lack of relaxation in the chemical environment? Specifically, how can we be certain that one reaction mechanism is more destabilized than another due to the absence of relaxation? In this regard, could the authors verify whether the algorithm converges on the same pathway when studying hydrogen diffusion (NH₂ hydrogenation) starting from reactants but with the chemical environment structure (frozen atoms) of the product?
2. Can the authors demonstrate the performance of the HDRL-FP framework for reactions that involve more than one atom diffusion? Could the authors please compute the optimal reaction pathway for a reaction that requires the active participation of more than one atom (e.g., the dissociation of H₂ on the surface in the H-B process)?
3. The authors have chosen a grid spacing of approximately 0.40 Å in each direction. I have concerns that this value might be relatively large considering chemical bonding distances. Have the authors tested the convergence of their mechanistic results with respect to this parameter?
4. Given that the studied reactions occur at high temperatures, anharmonic effects could significantly impact the computed free energy barriers. Could the authors please acknowledge this in the paper?

Reporting Summary

Nature Portfolio wishes to improve the reproducibility of the work that we publish. This form provides structure for consistency and transparency in reporting. For further information on Nature Portfolio policies, see our Editorial Policies and the Editorial Policy Checklist.

Statistics

For all statistical analyses, confirm that the following items are present in the figure legend, table legend, main text, or Methods section.

n/a Confirmed

- The exact sample size (n) for each experimental group/condition, given as a discrete number and unit of measurement
- A statement on whether measurements were taken from distinct samples or whether the same sample was measured repeatedly
- The statistical test(s) used AND whether they are one- or two-sided
Only common tests should be described solely by name; describe more complex techniques in the Methods section.
- A description of all covariates tested
- A description of any assumptions or corrections, such as tests of normality and adjustment for multiple comparisons
- A full description of the statistical parameters including central tendency (e.g. means) or other basic estimates (e.g. regression coefficient) AND variation (e.g. standard deviation) or associated estimates of uncertainty (e.g. confidence intervals)
- For null hypothesis testing, the test statistic (e.g. F , t , r) with confidence intervals, effect sizes, degrees of freedom and P value noted
Give P values as exact values whenever suitable.
- For Bayesian analysis, information on the choice of priors and Markov chain Monte Carlo settings
- For hierarchical and complex designs, identification of the appropriate level for tests and full reporting of outcomes
- Estimates of effect sizes (e.g. Cohen's d , Pearson's r), indicating how they were calculated

Our web collection on statistics for biologists contains articles on many of the points above.

Software and code

Policy information about availability of computer code

Data collection

We used Vienna Ab initio Simulation Package (VASP, with version 5.4) for density functional theory (DFT) calculations, VASP (v5.4) for nudged elastic band (NEB) and dimer calculations, and WarpDrive (v2.7) for reinforcement learning calculations. The multi-agent reinforcement learning code is explained in Ref. 22 and the WarpDrive (version 2.7) is available at <https://github.com/salesforce/warp-drive>. The environment code is available at https://github.com/salesforce/warp-drive/tree/master/example_envs.

Data analysis

In this study, we used VESTA (version 3.5.8) to analyze the data and results.

For manuscripts utilizing custom algorithms or software that are central to the research but not yet described in published literature, software must be made available to editors and reviewers. We strongly encourage code deposition in a community repository (e.g. GitHub). See the Nature Portfolio guidelines for submitting code & software for further information.

Data

Policy information about availability of data

All manuscripts must include a data availability statement. This statement should provide the following information, where applicable:

- Accession codes, unique identifiers, or web links for publicly available datasets
- A description of any restrictions on data availability
- For clinical datasets or third party data, please ensure that the statement adheres to our policy

All data generated in this study are provided in the Supplementary Information/Source Data file. The structures for DFT simulations are provided in the Supplementary Data file. These structures are visualized using Vesta (version 3.5.8) in this paper. Source data are provided with this paper.

Research involving human participants, their data, or biological material

Policy information about studies with human participants or human data. See also policy information about sex, gender (identity/presentation), and sexual orientation and race, ethnicity and racism.

Reporting on sex and gender	This study is not related to sex and gender since no human participants are involved in this study.
Reporting on race, ethnicity, or other socially relevant groupings	This study is not related to race, ethnicity or other socially related groupings since no human participants are involved in this study.
Population characteristics	This study is not related to population characteristics since no human participants are involved in this study.
Recruitment	This is not related since no human participants are involved in this study.
Ethics oversight	This is not related since no human participants are involved in this study.

Note that full information on the approval of the study protocol must also be provided in the manuscript.

Field-specific reporting

Please select the one below that is the best fit for your research. If you are not sure, read the appropriate sections before making your selection.

- Life sciences Behavioural & social sciences Ecological, evolutionary & environmental sciences

For a reference copy of the document with all sections, see [nature.com/documents/nr-reporting-summary-flat.pdf](https://www.nature.com/documents/nr-reporting-summary-flat.pdf)

Ecological, evolutionary & environmental sciences study design

All studies must disclose on these points even when the disclosure is negative.

Study description	This research focus on a reaction-agnostic framework based on high-throughput deep reinforcement learning with first principles (HDRL-FP) that offers excellent generalizability for investigating catalytic reactions. Using Haber-Bosch ammonia synthesis, we demonstrate the effectiveness of this framework.
Research sample	We focus on the hydrogen migration reaction of NH ₂ to NH ₃ on Fe surface since it is one of the most important reaction steps in Haber-Bosch reaction. We also examined the N ₂ diffusion on the Fe surface since they are related to important N ₂ adsorption/desorption steps. Our framework is demonstrated in both reactions. All the research are computational work.
Sampling strategy	We do the sampling when demonstrating the robustness and convergence of RL, as shown in Figure 2 and Figure 7, we select different numbers (e.g. n=4, 20, 100 and 500) of concurrent environment instances for the convergence of RL. These numbers were selected based on empirical testing. Our results (Figures 2 and 7) show that n=500 achieve good convergence within 60 mins computational time. For each environment instances (n), we did 5 independent runs to shows that the initial parameters of neural network do not affect the convergence. 5 runs were selected due to the reasonable standard deviation in both figure 2 and 6, especially for large n values.
Data collection	The data was collected from the computational simulations. The DFT simulations are not affected by any factors. The reinforcement learning simulations depend on the initial parameters of neural network. We did 5 independent simulations and demonstrate a good convergence (Figure 2 and 7) for large n values (> 20).
Timing and spatial scale	The data was collected when the simulations are done. The simulation results do not alter with time and spatial scale.
Data exclusions	No data was excluded from the analysis.
Reproducibility	All attempts to repeat the simulation results are successful. As mentioned above, the convergence time for RL depends on the initial parameters of neural network, but it converge well with large n values (>20).

Randomization

Blinding

Did the study involve field work? Yes No

Reporting for specific materials, systems and methods

We require information from authors about some types of materials, experimental systems and methods used in many studies. Here, indicate whether each material, system or method listed is relevant to your study. If you are not sure if a list item applies to your research, read the appropriate section before selecting a response.

Materials & experimental systems

n/a	Involvement in the study
[x]	[ ] Antibodies
[x]	[ ] Eukaryotic cell lines
[x]	[ ] Palaeontology and archaeology
[x]	[ ] Animals and other organisms
[x]	[ ] Clinical data
[x]	[ ] Dual use research of concern
[x]	[ ] Plants

Methods

n/a	Involvement in the study
[x]	[ ] ChIP-seq
[x]	[ ] Flow cytometry
[x]	[ ] MRI-based neuroimaging

Plants

Seed stocks

Novel plant genotypes

Authentication

REVIEWER COMMENTS

Reviewer #3 (Remarks to the Author):

The authors have mostly addressed my concerns, performing extra calculations in the process. Their responses and revisions are convincing and I recommend publishing this work in Nature Communications. In my view, further re-review is not needed.

Reviewer #4 (Remarks to the Author):

In this work, the authors describe the use of reinforcement learning (RL), a class of machine learning methods, for modeling chemical reactions at catalyst surfaces. I have been asked to comment on a revised version of the manuscript and, in particular, on the authors' response to the previous reviewers' comments.

In the previous round, Reviewer #1 gave a negative review on the grounds of judging the work as "incremental". Reviewer #2 was supportive of publication in principle but raised important technical questions. I will comment on the authors' replies individually, followed by some general remarks.

(1) Authors' response to Reviewer #1

The key criticism from Reviewer #1 is that they view the work as "incremental". They state that "the authors have published a few works on the same topic and it is such a big surprise that they still did not figure out the correct TS!" (p. 3 of the rebuttal letter). I feel that the reviewer's comment is perhaps unnecessarily hostile, and I do not think that the description of the transition state needs to prohibit publication, as long (!) as the authors can properly discuss the results.

I do nonetheless agree with the comment by Reviewer #1 that the authors have published on the same topic. The key references are two studies in JACS 2018 (10.1021/jacs.7b13409) and JACS 2021 (10.1021/jacs.1c08794). The former is a DFT-based study, using more conventional simulation techniques, and therefore does not affect novelty. The latter, however, bears rather strong similarity with the submitted work, and on this point I agree with Reviewer #1. In some parts this similarity is almost verbatim:

"Central to the DRL model is a policy $\pi\theta_p(at|st)$, represented by a deep neural network with parameters θ_p that analyzes the current state and takes actions. [...] The policy is then trained by a reinforcement learning strategy designed to maximize the expected value of the sum of discounted future rewards and conditioned on the success of reaching the final state." (p. 16806 of the JACS 2021 paper)

compared to

"Central to the deep RL model is a policy, denoted as $\pi\theta_p(at|st)$, which is represented by a deep neural network with parameters θ_p . This policy analyzes the current state and takes actions to the subsequent state. [...] The policy is subsequently trained via an RL strategy, aiming to maximize the expected sum of discounted future rewards, conditional on the success of reaching the final state." (p. 6 of the submitted manuscript)

I appreciate that descriptions of the methods are sometimes similar between papers from the same authors, but this text is given in the "Results and Discussion" sections in both cases, which could create the impression that the deep RL approach is used here for the first time. No reference to the JACS paper is made in the paragraph cited above. The text overlap above is also (in my mind) inconsistent with the authors' statement in the rebuttal letter that

"[their] previous publications on the H-B reaction mechanisms predominantly focused on DFT studies, constructing reaction pathways based on the combination of chemical intuition and DFT simulations. The current work is distinctly different. This work innovatively leverages the frontier of most advanced AI techniques to the study of catalytic reactions." (p. 3 of the response letter)

It would be very important to make much clearer in what way the submitted work is "distinctly different" from the JACS 2021 paper, which also used "advanced AI techniques" (by which I assume the authors mean the use of RL).

Reviewer #1 also comments that "the model is also quite simple", e.g. with regard to keeping the surface fixed (p. 2 of the rebuttal letter). I feel that the authors have addressed this point appropriately with the added Figure S4, which in fact could be a main text figure.

(2) Authors' response to Reviewer #2

Reviewer #2, whilst overall positive, raised a number of technical comments that I think are important.

- Comment #2.1 asks to repeat one of the simulations with the product geometry to rule out major effects of the structure. This is a valid request and I think the authors would need to either address it or justify why they do not do so. At the moment, the authors merely state that "While we acknowledge the referee's suggestion to perform NH₂ hydrogenation under different environments, we opted for another calculation incorporating structure relaxation", without giving reasons. I do not think that this is a satisfactory response.

- Comment #2.2 asks to address a situation where more than one atom is involved in the reaction, giving a specific example that the reviewer would like to see. The authors have added N₂ diffusion, which does show an extension of the scope, but I do not think that it provides what Reviewer #2 asked for. Again, this would be important to include for a convincing response. If the method is not capable of doing so, this need not stand in the way of publication, but it would need to be addressed.

- Comment #2.3 is related to the grid spacing which the species are moved on (approximately 0.4 Å). I share Reviewer #2's view that these are relatively coarse grids. For example, when calculating bond dissociation curves of molecules, one might typically expect to see at least 0.1 Å spacings. The authors state in the revised SI that "it is important to note that increasing the density of grid points substantially increases computational demands [...] [which] does not necessarily translate to a proportional improvement in the effectiveness of the reinforcement learning search" (p. S2). I think that this statement would need to be supported by quantitative results (e.g. re-running the simulations with a 0.2 Å grid spacing, which I believe is what Reviewer #2 would like to see here).

- Comment #2.4 is related to anharmonic effects and I feel that this comment has been satisfactorily addressed.

(3) General comments

In addition to the above, I think that the presentation could be improved, especially when submitting to a high-ranking journal. Some suggestions:

- The authors state in the rebuttal letter that they "discuss these challenges in [their] Supplemental Information (SI) Section 1, titled "Challenges of Using Reinforcement Learning in Chemical Reactions."" (p. 1). Would it not be worth discussing some of these challenges in the Introduction?

- The authors should keep in mind that Nature Communications has a wider-ranging audience than JACS (i.e. it includes many readers who are not chemists). It would be good to start, for example, by showing the key reaction pathways early on. The current Figure 1 is rather general and could

be improved by making it clearer what the domain problem for the present work is (Figure 4 could be useful here!).

- It may not be obvious to the reader what Figures 2 and 6 show and it would be helpful to explain this more clearly in the captions. What does the color coding indicate? Can this be replaced with a sequential colormap?

Finally, a data availability statement is missing from the paper and it would be important to give assurance that all data (including structures) and code required to reproduce the work will be made available to the reader. This is consistent with common practice in the field and with the policies of Nature family journals.

Manuscript ID: NCOMMS-23-37115A

Title: Massively High-Throughput Deep Reinforcement Learning with First Principles: A Generalizable Approach to Investigating Catalytic Reaction Mechanisms

Authors: Tian Lan, Huan Wang, and Qi An

Response to reviewer 3.

Thanks for the comments. We have reproduced the comments *in italics* below. **Our responses** are in **bold**.

Comments:

The authors have mostly addressed my concerns, performing extra calculations in the process. Their responses and revisions are convincing and I recommend publishing this work in Nature Communications. In my view, further re-review is not needed.

Authors Reply:

We thank the referee for the positive comments and recommendation for publication.

Manuscript ID: NCOMMS-23-37115A

Title: Massively High-Throughput Deep Reinforcement Learning with First Principles: A Generalizable Approach to Investigating Catalytic Reaction Mechanisms

Authors: Tian Lan, Huan Wang, and Qi An

Response to reviewer 4.

Thanks for the comments. We have reproduced the comments *in italics* below. **Our responses** are in **bold**.

Comments:

In this work, the authors describe the use of reinforcement learning (RL), a class of machine learning methods, for modeling chemical reactions at catalyst surfaces. I have been asked to comment on a revised version of the manuscript and, in particular, on the authors' response to the previous reviewers' comments. In the previous round, Reviewer #1 gave a negative review on the grounds of judging the work as "incremental". Reviewer #2 was supportive of publication in principle but raised important technical questions. I will comment on the authors' replies individually, followed by some general remarks.

(1) Authors' response to Reviewer #1

The key criticism from Reviewer #1 is that they view the work as "incremental". They state that "the authors have published a few works on the same topic and it is such a big surprise that they still did not figure out the correct TS!" (p. 3 of the rebuttal letter). I feel that the reviewer's comment is perhaps unnecessarily hostile, and I do not think that the description of the transition state needs to prohibit publication, as long (!) as the authors can properly discuss the results.

Authors Reply:

We thank the referee for pointing out the "unnecessarily hostile" comments from the reviewer #1.

I do nonetheless agree with the comment by Reviewer #1 that the authors have published on the same topic. The key references are two studies in JACS 2018 (10.1021/jacs.7b13409) and JACS 2021 (10.1021/jacs.1c08794). The former is a DFT-based study, using more conventional simulation techniques, and therefore does not affect novelty.

Authors Reply:

We thank the referee for pointing out that the JACS 2018 (10.1021/jacs.7b13409) paper does not affect the novelty of the current manuscript.

The latter, however, bears rather strong similarity with the submitted work, and on this point I agree with Reviewer #1. In some parts this similarity is almost verbatim:

“Central to the DRL model is a policy $\pi_{\theta p}(a_t|s_t)$, represented by a deep neural network with parameters θp that analyzes the current state and takes actions. [...] The policy is then trained by a reinforcement learning strategy designed to maximize the expected value of the sum of discounted future rewards and conditioned on the success of reaching the final state.” (p. 16806 of the JACS 2021 paper)

compared to

“Central to the deep RL model is a policy, denoted as $\pi_{\theta p}(a_t|s_t)$, which is represented by a deep neural network with parameters θp . This policy analyzes the current state and takes actions to the subsequent state. [...] The policy is subsequently trained via an RL strategy, aiming to maximize the expected sum of discounted future rewards, conditional on the success of reaching the final state.” (p. 6 of the submitted manuscript)

I appreciate that descriptions of the methods are sometimes similar between papers from the same authors, but this text is given in the “Results and Discussion” sections in both cases, which could create the impression that the deep RL approach is used here for the first time. No reference to the JACS paper is made in the paragraph cited above. The text overlap above is also (in my mind) inconsistent with the authors’ statement in the rebuttal letter that “[their] previous publications on the H-B reaction mechanisms predominantly focused on DFT studies, constructing reaction pathways based on the combination of chemical intuition and DFT simulations. The current work is distinctly different. This work innovatively leverages the frontier of most advanced AI techniques to the study of catalytic reactions.” (p. 3 of the response letter)

It would be very important to make much clearer in what way the submitted work is “distinctly different” from the JACS 2021 paper, which also used “advanced AI techniques” (by which I assume the authors mean the use of RL).

Authors Reply:

Concerning the novelty of our current work and its fundamental distinctions from prior research (JACS 2021 (10.1021/jacs.1c08794)), we offer discussions in Introduction. However, we recognize the possibility that our explanations may lack clarity for the referee and other readers. Additionally, we acknowledge the referee's valid point regarding the potential confusion stemming from the policy definition in the "Results and Discussion" section, which reads similar to the policy definition in the JACS 2021 paper.

Hence, we kindly appreciate the opportunity to elaborate this further.

1. The primary issue raised by the referee pertains to the perceived resemblance of this paragraph to our earlier publication:

“Central to the deep RL model is a policy, denoted as $\pi_{\theta p}(a_t|s_t)$, which is represented by a deep neural network with parameters θp . This policy analyzes the current state and takes actions to the subsequent state. [...] The policy is subsequently trained via an RL strategy, aiming to maximize the expected sum of discounted future rewards, conditional on the success of reaching the final state.

This paragraph aims to provide a textbook definition of the reinforcement learning policy, presenting the description in a general manner. This explains the similarity between the definitions of our current and previous work. Recognizing that placing this paragraph in the Discussion section is indeed misleading, we have relocated it to the Method section (p. 21 of the revised manuscript) while integrating the key definition of RL into page 5 of the revised manuscript.

2. Novelties and Innovations of this work:

In this work, we present two significant innovations: the introduction of a generalizable RL representation for chemical reactions, and the development of a massively high-throughput computational framework to support this RL simulation.

2.1 Generalizable RL Environment Representation for Chemical Reactions

A significant challenge, and simultaneously one of the primary innovations, lies in defining and modeling the "state" and the "action" - the core elements in the policy $\pi(action|state)$ that shape the representation of the RL environment. It is worth noting that the environment design, including the specification of states and actions, is fundamental to the success of reinforcement learning. A carefully crafted environment facilitates effective learning, generalization, and transferability of knowledge, ultimately leading to the development of robust and adaptive RL agents capable of solving complex problems. For example, one major novelty in the famous AlphaGo paper (Nature, vol. 529 (2016), pp 484) lies in the use of the raw image of board positions as states. This approach allowed AlphaGo to capture complex spatial patterns and dependencies inherent in the game. It represented a departure from traditional handcrafted feature representations and paved the way for significant advancements in computer Go playing strength.

For our problem, delving into the intricate complexities of investigating catalytic reaction mechanisms presents an even more formidable obstacle to the broad application of reinforcement learning across various catalytic reactions. Consequently, prior research (JACS 2021 (10.1021/jacs.1c08794) on applying reinforcement learning to chemical reactions has predominantly concentrated on modeling the "states" and "actions" through a simplified or semi-empirical representation based on specific chemical reactions. These methodologies heavily rely on the laborious design of specialized state vector encodings, heuristic action guidelines, or nonlinearly transforming reward functions. Such an approach can severely constrain the adaptability of the reinforcement learning algorithm and confine exploration to a predetermined set of reaction networks, thus impeding the discovery of novel elementary reaction mechanisms.

To understand the novel contribution of our work, let's provide a detailed comparison of the fundamental disparities in "states" and "actions" (and thereby the representation of the environment) between our current work and previous JACS paper (10.1021/jacs.1c08794). We will elucidate our "generalizable approach" introduced in this study and its innovative

aspects, which effectively address the fundamental limitations encountered in previous reinforcement learning research pertaining to chemical reactions.

States:

a. Current (generalizable approach):

We define the reaction-agnostic states by simply using the Cartesian coordinates of atom positions. Consequently, the states are represented by the normalized coordinates of the migrating atom and its Euclidean distance to the target position. For multiple migrating atoms, the states will be defined as the concatenated coordinates of these atoms normalized individually.

b. Previous (10.1021/jacs.1c08794):

Rather than delving into the intricacies of individual atoms, the previous approach confines states to a predetermined set of network configurations specific to a reaction.

For example, in the case of the Fe-bcc (111) surface of the H-B process studies at JACS 2021 paper, we introduce an encoded state vector consisting of 23 elements, accommodating a total of 20 surface sites within it. The value of each entry represents the adsorbents and the adsorbents H, N, NH, NH₂, NH₃, and N₂ are encoded by integers 1–6, respectively, and an empty site is represented by the number 0. The last three entries of the vector are adopted to record the number of gas species (N₂↔21, H₂↔22, and NH₃↔23) along the reaction path. For example, the initial state starting from the reactant (4N surface configuration with three H₂ and one N₂ in the gas phase) is represented by the state vector [00002222000000000000130]

Actions:

a. Current (generalizable approach):

We define the actions as stepwise movement of the migrating atom in six possible directions within the 3-dimensional grid: forward, backward, up, down, left, and right. For various atom actors, we have devised a two-dimensional action space consisting of “atom choice” and “move direction”. For example, the action (1, down) means that the first atom actor moves downward.

b. Previous:

In the case of the Fe-bcc (111) surface of the H-B process studies at JACS 2021 paper, the actions connecting states are defined in terms of four main categories: (1) H₂ adsorption/desorption from the surface; (2) N₂ adsorption, migration, and dissociation on the surface; (3) surface hydrogen migration to hydrogenate N, NH, or NH₂ species (for example H + NH ⇒ NH₂); and (4) NH₃ desorption from the surface. Based on the specific surface sites

for reactions, the actions are further refined to form the action space utilized in the environment. The definitions of all 17 actions represent the reaction steps known on the Fe-bcc (111) surface and are displayed in Figures S3–S6 of the SI in JACS 2021 paper.

Upon comparing our current approach with the previous one, it becomes apparent that our work presents a novel and generalizable RL representation of catalytic reactions solely reliant on the Cartesian coordinates of atoms. Therefore, our method is reaction-agnostic, circumventing the laborious and un-scalable semi-empirical modeling of reaction-specific environments employed in previous methods. It is worth noting that the previous approach also requires a substantial amount of domain knowledge and confines exploration to a predetermined set of reaction networks, thus impeding the discovery of unknown elementary reaction mechanisms.

2.2 Massively High-Throughput Computation

Our approach, featuring a generalizable representation of states and actions that goes beyond specific reactions, is bolstered by our evolutionary framework, HDRL-FP, for high-throughput (supporting thousands of concurrent RL environments) and low-cost computations (running on a single GPU). This novel framework significantly diminishes correlations and noise between exploration steps in RL, ensuring training stability, accelerating convergence, and mitigating the need for fine-tuning hyperparameters. We have detailed discussions of this engineering breakthrough in Section 2.1.3: Massively High-Throughput Framework.

As discussed, without our highly efficient computational architecture for high-throughput processing, RL applied to a generalizable environment representation based on atomic coordinates would struggle to converge due to its computational complexity of chemical reactions featuring high noise, strong correlation and non-stationarity.

To comprehend the pivotal role of our computational architecture, we conducted quantitative studies on RL simulations operating with varying numbers of concurrent environments. Both Fig. 2 and Fig. 6 demonstrate that, when utilizing consistent fixed hyperparameters, simulations with an increased number of concurrent environments achieve global convergence faster and more reliably. Building upon these quantitative findings, Section 2.2 in our manuscript provides detailed discussions on the crucial role of massive parallelism in facilitating effective RL exploration of a wide array of reaction mechanisms through a generalizable environment representation constructed solely from atomic positions.

Furthermore, this addresses another query posed by the referee regarding the significance of Fig. 2 and 6. Each curve in the figures illustrates accumulated reward (Eq. 1) plotted against training wall-clock time, a conventional metric in RL for gauging learning speed and stability (Reference: *Mnih, V. et al. Asynchronous methods for deep reinforcement learning. Proc. 528 Machine Learning Res. 48, 1928–1937 (2016).*). Distinct colors denote varying levels of computational concurrencies, while the shaded regions depict the error bars derived from five independent measurements.

To clarify this, in addition to our previous discussion, we added the discussion below on page 3 in the revised manuscript.

“For instance, our prior RL study¹⁵ developed an encoded state vector consisting of 23 elements, accommodating a total of 20 surface sites within the catalytic surface. The value of each entry represents various adsorbents and are encoded by the integers. In addition, the actions connecting states are defined in terms of some specific chemical reactions in which the atomic motion are not explicitly involved in these reactions. This type of RL application heavily depends on the domain knowledge and lacks generality when using it in other complex reactions. Therefore, it is limited to managing intricate reaction paths defined by a predetermined assortment of diverse reaction mechanisms. Meanwhile, the elementary task of probing undiscovered reaction mechanisms remains elusive.”

Reviewer #1 also comments that “the model is also quite simple”, e.g. with regard to keeping the surface fixed (p. 2 of the rebuttal letter). I feel that the authors have addressed this point appropriately with the added Figure S4, which in fact could be a main text figure.

Authors Reply:

We thank the referee for the comment and suggestion. We have moved Figure S4 to the main text as Figure 6.

(2) Authors' response to Reviewer #2

Reviewer #2, whilst overall positive, raised a number of technical comments that I think are important.

- Comment #2.1 asks to repeat one of the simulations with the product geometry to rule out major effects of the structure. This is a valid request and I think the authors would need to either address it or justify why they do not do so. At the moment, the authors merely state that "While we acknowledge the referee's suggestion to perform NH₂ hydrogenation under different environments, we opted for another calculation incorporating structure relaxation", without giving reasons. I do not think that this is a satisfactory response.

Authors Reply:

We agree with the referee that we need to provide the reason that we select to study the N diffusion instead of NH₂ hydrogenation to test our HDRL-FP framework under a different chemical environment (non-relax vs. relaxation environment).

First, it is essential to underscore the problem that our research seeks to tackle through the utilization of the HDRL-FP framework. Given our knowledge of the initial reactant and final product configurations, our goal is to employ HDRL-FP to search for the reaction pathway. For the case of NH₂ hydrogenation, allowing relaxation may significantly affect the configurations of initial reactant, intermediates states and final products. The H migration

process involves the movement of H across discrete mesh points between the initial and final states. As the migrating H approaches the Fe surface closely, it would repel adjacent Fe atoms, potentially altering or compromising the Fe(111) surface. To preserve these configurations, we must impose specific constraints within our HDRL-FP calculations. However, defining constraints for the NH₂ hydrogenation reaction presents a greater challenge. For instance, in the ER mechanism for NH₂ hydrogenation, it is essential for the initial H atom to be positioned away from the surface. Allowing relaxation perpendicular to the surface could cause the H atom move towards the surface, thereby potentially altering the configuration of the initial reactant significantly.

Consequently, we decided to use the N diffusion to illustrate that our HDRL-FP framework works well under a different chemical environment. In our examination of N diffusion within a relaxed environment, we allowed for the relaxation of migrated N in a direction perpendicular to the surface. Indeed, this relaxation makes the potential energy surface (PES) smoother compared to the non-relaxed environments. However, the overall shape of the PES remains consistent with the non-relaxed environments, and the predicted reaction paths are identical for both environments. To clarify this, we have added the following sentences on page 18 of the revised manuscript.

“Investigating the performance of our HDRL-FP framework under varying chemical environments is crucial, especially considering the environments in which atoms are allowed to relax. It is important to note that maintaining the configurations of the initial reactants and final products necessitates the implementation of certain constraints. Therefore, we have employed N diffusion as an example to examine this aspect.”

- Comment #2.2 asks to address a situation where more than one atom is involved in the reaction, giving a specific example that the reviewer would like to see. The authors have added N₂ diffusion, which does show an extension of the scope, but I do not think that it provides what Reviewer #2 asked for. Again, this would be important to include for a convincing response. If the method is not capable of doing so, this need not stand in the way of publication, but it would need to be addressed.

Authors Reply:

This comment aims to evaluate if our Reinforcement Learning (RL) model is adaptable to systems comprising two atoms. The suggestion by the referee to explore hydrogen (H₂) may stem from our current focus on hydrogen migration reactions. However, H₂ dissociation does not play a significant role in the H-B reaction. In our prior research (JACS 2019), we found that the barrier for H₂ dissociative chemisorption is sufficiently small to have negligible effects on kinetics. Therefore, the H₂ dissociation is not considered as the main reaction steps in the free energy landscape (Figure 5 of the manuscript). In contrast, our attention shifts to a nitrogen (N₂) migration for the two-atom scenario, involving multiple reaction steps and representing one of the crucial rate-determining steps. Thus, we exemplify our RL model's applicability to two-atom systems through the transition from γ -N₂ configuration to δ -N₂ configuration among the N₂ related reactions.

It is worth noting that based upon our novel action design, our method is scalable and generalizable to any multi-atom system, as we elaborated in the “action” definition in the

Section “Generalizable RL Environment Representation for Chemical Reactions” in our response, as well as in the manuscript. Specifically, we have developed a two-dimensional action space that incorporates both "atom choice" and "move direction" simultaneously. This innovative design has proven to be highly effective in our investigation of N₂ migration for the two-atom scenario. Given that the primary focus of the current manuscript is to establish the method and concentrate on reactions related to single-atom motion, we will defer the investigation of other N₂-related reaction steps to future research.

To clarify this, we added the sentences below on page 20 of the revised manuscript.

“N₂ migration for the two-atom scenario involves multiple reaction steps and represents one of the crucial rate-determining steps.²⁴ Thus, we demonstrate our HDRL-FP model's applicability to two-atom systems through the transition from γ -N₂ configuration to δ -N₂ configuration among the N₂ related reactions. Given that the primary focus of the current manuscript is to establish the method and concentrate on reactions related to single-atom motion, we will defer the investigation of other N₂-related reaction steps to future research.”

- Comment #2.3 is related to the grid spacing which the species are moved on (approximately 0.4 Å). I share Reviewer #2's view that these are relatively coarse grids. For example, when calculating bond dissociation curves of molecules, one might typically expect to see at least 0.1 Å spacings. The authors state in the revised SI that "it is important to note that increasing the density of grid points substantially increases computational demands [...] [which] does not necessarily translate to a proportional improvement in the effectiveness of the reinforcement learning search" (p. S2). I think that this statement would need to be supported by quantitative results (e.g. re-running the simulations with a 0.2 Å grid spacing, which I believe is what Reviewer #2 would like to see here).

Authors Reply:

We thank the referee for these insightful comments. Although we believe that the grid spacing of 0.4 Å is adequate to represent the important aspects of the potential energy surface for this study, we follow the suggestion from the referee and reran the simulation with a 0.2 Å grid spacing. It is important to note that this adjustment effectively increased the grid density by eightfold. Given that each atom has six different action choices at each grid vertex, this leads to an exponential increase in the number of possible paths, roughly by a factor of 6^M, where M could be close to 7, signifying a significant increase. While the complexity can be mitigated by selectively increasing density in certain regions, or by resuming finer exploration from the previous learnt policy tackling the 0.4 Å scenario, we opted to challenge our framework and test it under the highest level of complexity for this problem. Consequently, we uniformly increased the grid density from 0.4 Å spacing to 0.2 Å, and asked RL to directly explore this new environment from scratch.

We reran the small grid calculation for the reaction of NH₂ hydrogenation, and we explore two potential reaction mechanisms: (1) the Langmuir-Hinshelwood (LH) mechanism, where both reactants are present on the catalytic surface, and (2) the Eley-Rideal (ER) mechanism, where one hydrogen atom originates from the gas phase. Our DFT calculations show that 0.4 Å grid spacing generates the smooth potential energy surface (PES) along the reaction pathway, as evidenced by the absence of significant energy fluctuations between adjacent

grid points (Figure 3 of the manuscript). Therefore, we generate the energetic data for 0.2 Å grid based on the interpolation of 0.4 Å DFT data. To perform uniform interpolation for the mesh grid, transitioning from 0.4 Å to 0.2 Å, we employ proximal interpolation based on the nearest neighbor method, commonly utilized in 3D rendering to determine color values for textured surfaces. In our scenario, three categories of empty sites necessitate interpolation: (1) Midpoints of grid edges (edge center of two neighboring points of 0.4 Å gridding): We compute the average value of the two vertices comprising the edge; (2) Centers of grid surfaces (face center of 4 neighboring points of 0.4 Å gridding): We calculate the average value of the four vertices composing each surface; and (3) Centers of grid cubic cells (body center of 8 neighboring grids of 0.4 Å gridding): We determine the average value of the eight vertices constituting each cell.

To demonstrate that the interpolation provides an accurate description of PES, we did the DFT calculations along both ER and LH reaction paths. As shown in Figure 1 below, the interpolation along both reaction paths agrees very well with the DFT calculations. Therefore, our new RL calculations of 0.2 Å grid are based on this new PES data.

Figure 1. The comparison between the direct DFT simulations for 0.2 Å grid spacing and the data from interpolation based on 0.4 Å grid spacing DFT: (a) the reaction path for the ER mechanism of H migration to NH_2 , forming NH_3 ; (b) the reaction path for LH mechanism of H migration to NH_2 , forming NH_3 . For the orange curves, half of the data are DFT (even steps) and half of the data are from the interpolation (odd steps), as discussed in the main text.

Figures 2 (a, b) illustrate the convergence speed when processing the LH and ER reactions, respectively, with 500 environment replicas running in parallel. Our findings reveal that in all separate runs, LH simulations reach the same global optimum within 135 minutes (versus 45 minutes under the 0.4 Å setting), whereas ER simulations achieve the global optimum within 90 minutes (versus 40 minutes under the 0.4 Å setting). These experiments once again underscore the robustness and consistent convergence of our method. The convergence of RL is slower compared to previous calculations of 0.4 Å grid spacing, owing to the exponential increase of the complexity. However, the obtained reaction paths for both ER

and LH mechanisms are consistent with the prediction of 0.4 Å grid spacing. These new calculations demonstrate that our previous calculation with 0.4 Å grid spacing obtains reliable information with significantly reduced computational cost.

Figure 2: Convergence and learning speed, measured in total runtime using wall-clock minutes, for our framework with 500 environment replicas under 0.2 Å setting applied to (a) Langmuir-Hinshelwood and (b) Eley-Rideal hydrogenation reaction of NH₂ to NH₃. The episodic reward is the mean accumulated reward that H atom actors collect from the initial to the terminal state of (a) Langmuir-Hinshelwood and (b) Eley-Rideal. For robustness, the depicted results are averaging over five independent runs from scratch with different initialization seeds and the same hyperparameters. The shadow regions represent the error bar of five independent runs.

To clarify this, we add the new calculations of 0.2 Å grid spacing to section 6 of the supporting information. We also added the sentence below on page 14 to discuss the calculations of 0.2 and 0.4 griding space.

“It is worth noting that the HDRL-FP calculations are based on a mesh grid of 0.4 Å in the supercell. The convergence of the griding space has been demonstrated by a more complex calculations with 0.2 Å grid spacing. The details of these calculations are discussed in the SI and Figure S2 and Figure S3 of SI.”

- Comment #2.4 is related to anharmonic effects and I feel that this comment has been satisfactorily addressed.

Authors Reply:

We thank the referee for this positive comment.

(3) General comments

In addition to the above, I think that the presentation could be improved, especially when submitting to a high-ranking journal. Some suggestions:

- The authors state in the rebuttal letter that they "discuss these challenges in [their] Supplemental Information (SI) Section 1, titled "Challenges of Using Reinforcement Learning in Chemical Reactions."" (p. 1). Would it not be worth discussing some of these challenges in the Introduction?

Authors Reply:

We thank the referee for the comments, and we have added the discussion below in the introduction section of the revised manuscript.

“Deep reinforcement learning (RL) is heralded as a paradigmatic approach for automating the exploration of unknown reaction mechanisms in a first-principles manner. However, RL faces many scientific and engineering challenges in its practical application to this task. For instance, issues such as non-stationarity, substantial bias, and strong correlation in data sequences commonly found in catalytic reaction pathways can impede the policy exploration in RL. This impediment can prevent effective escape from local minima, leading to impossible or excessively slow convergence.”

- The authors should keep in mind that Nature Communications has a wider-ranging audience than JACS (i.e. it includes many readers who are not chemists). It would be good to start, for example, by showing the key reaction pathways early on. The current Figure 1 is rather general and could be improved by making it clearer what the domain problem for the present work is (Figure 4 could be useful here!).

Authors Reply:

We thank the referee for the suggestion and agree that Nature Communications has a wider-ranging audience than JACS. In the revised manuscript, we have modified Figure 1 by adding some images of Figure 4. We also added the sentences below on page 9 in the revised manuscript to clarify the domain problem we plan to solve:

“Employing this framework, we aim to address the generalized and complex chemical reactions, as shown in Figure 1(b), in which the atomic motions are involved explicitly.”

- It may not be obvious to the reader what Figures 2 and 6 show and it would be helpful to explain this more clearly in the captions. What does the color coding indicate? Can this be replaced with a sequential colormap?

Authors Reply:

We have mentioned this in our previous response and would like to repeat the response here. Each curve in the figures illustrates accumulated reward (Eq. 1) plotted against training wall-clock time, a conventional metric in RL for gauging learning speed and stability (Reference: Mnih, V. et al. *Asynchronous methods for deep reinforcement learning. Proc. 528 Machine Learning Res. 48, 1928–1937 (2016).*). Distinct colors denote varying levels of computational concurrencies, while the shaded regions depict the error bars derived from five independent measurements. Since the color regions represent the error bar, it cannot be replaced with sequential colormap. The partial overlay of color regions may be unavoidable due to the potential overlap of error bars from different curves. The explanations for the color coding are in the captions of Figures 2/6 and are also copied here.

“Different numbers of concurrent environment instances were used - 4 in red, 20 in green, 100 in blue, and 500 in yellow. The shadow regions represent the error bar of five independent runs.”

Finally, a data availability statement is missing from the paper and it would be important to give assurance that all data (including structures) and code required to reproduce the work will be made available to the reader. This is consistent with common practice in the field and with the policies of Nature family journals.

Authors Reply:

We thank the referee for these comments. Indeed, the data availability statement and code availability statement are in our manuscript (before references section). We are happy to share our research results to readers after the manuscript is published.

REVIEWERS' COMMENTS

Reviewer #4 (Remarks to the Author):

The authors have addressed most of my concerns in the revised submission. There are a few points that I feel require further clarification before proceeding further.

(1) I very much like the authors' discussion on pp. 3-7 of the rebuttal, which now makes it much clearer than before how their new submission is different from their 2021 JACS paper. It would be helpful to bring as much as possible of this discussion into the main text (in addition to the changes made to the Introduction), rather than "just" the rebuttal letter - for example, it could be incorporated into the Discussion section at the end. The difference between the existing approach and the new one would also be helpful to illustrate in Fig. 1b (at the moment, this panel is only very briefly discussed and the caption does not explain the various components and abbreviations).

(2) The responses to #2.1 and #2.2 (pp. 7-9 of the rebuttal) are relatively general. Essentially, the authors are saying that they have to defer the study of these reactions to future work. This is fine, given the new aspects discussed in (1) above, although I think that the concluding statement on p. 22 that "Therefore HDRL-FP can adapt to a multitude of catalytic reactions" needs to be tuned down. This paragraph would benefit from a statement of what the authors' approach currently cannot do, and what expected next steps are.

(3) Regarding my comments on p. 12 of the rebuttal - I think that both the added discussion of challenges of RL in chemical reactions, and the discussion of the new Fig. 1b, are quite brief and generic. It would help the manuscript to be more specific here, and in particular to use the Introduction to describe more clearly the difference between previous work and this one, as mentioned above. This is a relatively new technique that will likely be interesting to many readers in chemistry, so I think it is worth improving the presentation here.

(4) On p. 13, I think there was a misunderstanding regarding the "colormap". What I suggest is to replace the categorical color-coding of red -> green -> blue -> yellow with sequential colors. For example, the color coding could run from light blue (4) to dark blue (500) in Fig. 2 and 7. This would make it clearer that the authors are gradually increasing the number of concurrent instances.

(5) Related to a point in my previous report, it would be good to see a more helpful data and code availability statement. For example, the coordinate files for the structures are not provided, and these together with the code would allow a reader to reproduce the findings of the work (and ideally to help them to get started modeling other catalytic reactions!). It would also be good to provide the actual code used to train the models, for example in a Jupyter notebook, and the model parameters. The repository link currently refers to the general RL framework, but does not seem to include the particular code used in this work.

Manuscript ID: NCOMMS-23-37115B

Title: Massively High-Throughput Deep Reinforcement Learning with First Principles: A Generalizable Approach to Investigating Catalytic Reaction Mechanisms

Authors: Tian Lan, Huan Wang, and Qi An

Response to reviewer 4.

Thanks for the positive comments. We have reproduced the comments *in italics* below. **Our responses are in bold.**

Comments:

The authors have addressed most of my concerns in the revised submission. There are a few points that I feel require further clarification before proceeding further.

(1) I very much like the authors' discussion on pp. 3-7 of the rebuttal, which now makes it much clearer than before how their new submission is different from their 2021 JACS paper. It would be helpful to bring as much as possible of this discussion into the main text (in addition to the changes made to the Introduction), rather than "just" the rebuttal letter - for example, it could be incorporated into the Discussion section at the end. The difference between the existing approach and the new one would also be helpful to illustrate in Fig. 1b (at the moment, this panel is only very briefly discussed and the caption does not explain the various components and abbreviations).

Authors Reply:

We agree with the referee and add new discussions on page 23 in the revised manuscript.

2.7 Advances in Generalizable RL Framework for Chemical Simulations

“In this HDRL-FP framework, two significant advances are present: the introduction of a generalizable RL representation for chemical reactions and the development of a high-throughput strategy to support this RL simulation. One of the primary differences between this work and previous RL method¹⁵ lies in defining and modeling the "state" and "action," which are crucial elements in the policy $\pi(\text{action}|\text{state})$ that shape the RL environment. Effective environment design, including the specification of states and actions, is fundamental to the success of RL. A well-crafted environment facilitates effective learning, generalization, and transferability of knowledge, ultimately leading to robust and adaptive RL agents capable of solving complex problems. For example, a major progress in the famous AlphaGo work was using raw board position images as states, allowing AlphaGo to capture complex spatial patterns and dependencies inherent in the game.¹²

The complexities of investigating catalytic reaction mechanisms present a formidable obstacle to the broad application of RL across various catalytic reactions. Prior research on applying RL to chemical reactions has predominantly focused on modeling states and actions through a simplified or semi-empirical representation specific to certain chemical reactions.^{15,16} For example, in our previous study,¹⁵ we introduced an encoded state vector consisting of 23 elements, accommodating a total of 20 surface sites and three gas species. To address these limitations, our work here introduces a generalizable approach that defines

reaction-agnostic states using the Cartesian coordinates of atom positions. States are represented by the normalized coordinates of the migrating atom and its Euclidean distance to the target position. Actions are defined as stepwise movements of the migrating atom in six possible directions within a 3-dimensional grid. This method avoids the laborious and unscalable semi-empirical modeling of reaction-specific environments employed in previous methods.

Our approach, featuring a generalizable representation of states and actions, is boosted by our evolutionary framework, HDRL-FP, for high-throughput (supporting thousands of concurrent RL environments) and low-cost computations (running on a single GPU). This framework significantly diminishes correlations and noise between exploration steps in RL, ensuring training stability, accelerating convergence, and mitigating the need for fine-tuning hyperparameters. Without this efficient computational architecture for high-throughput processing, RL applied to a generalizable environment representation based on atomic coordinates would struggle to converge due to its computational complexity, high noise, strong correlation, and non-stationarity of chemical reactions. Quantitative studies on RL simulations operating with varying numbers of concurrent environments show that simulations with an increased number of concurrent environments achieve global convergence faster and more reliably. This highlights the crucial role of massive parallelism in facilitating effective RL exploration of a wide array of reaction mechanisms through a generalizable environment representation constructed solely from atomic positions.”

(2) The responses to #2.1 and #2.2 (pp. 7-9 of the rebuttal) are relatively general. Essentially, the authors are saying that they have to defer the study of these reactions to future work. This is fine, given the new aspects discussed in (1) above, although I think that the concluding statement on p. 22 that "Therefore HDRL-FP can adapt to a multitude of catalytic reactions" needs to be tuned down. This paragraph would benefit from a statement of what the authors' approach currently cannot do, and what expected next steps are.

Authors Reply:

We agree with the referee that our HRRL-FP approach needs to be combined with proper design of constraints when studying some chemical reactions in the relaxation environments. To clarify this, on P22-23, we revised the sentence below from

“Therefore HDRL-FP can adapt to a multitude of catalytic reactions”

To:

“HDRL-FP can adapt to a wide range of catalytic reactions depending on the complexity of the environments. Our current approach works well for both non-relaxation environments and relaxation environments with well-defined constraints (e.g., relaxing along one direction). Combining HDRL-FP with proper constraint design will be an important future research direction for exploring catalytic reaction mechanisms in complex environments.”

(3) Regarding my comments on p. 12 of the rebuttal - I think that both the added discussion of challenges of RL in chemical reactions, and the discussion of the new Fig. 1b, are quite brief and generic. It would help the manuscript to be more specific here, and in particular to use the

Introduction to describe more clearly the difference between previous work and this one, as mentioned above. This is a relatively new technique that will likely be interesting to many readers in chemistry, so I think it is worth improving the presentation here.

Authors Reply:

We have added a new discussion section to clarify the difference between previous work and this one. But we agree with the referee that we need to clarify Figure 1b and the challenges of RL in chemical reactions. Therefore, we added a few sentences below on page xx in the revised manuscript.

To clarify the challenges of RL, we move section 1 in SI to the main text.

“Although deep reinforcement learning (RL) is considered as one ultimate epitome of exploring unknown reaction mechanisms in an automated and first-principles way, there are many scientific and engineering challenges to the use of RL for this difficult task. For example, in online RL, the sequence of data observed by the learning agent could be non-stationary and strongly correlated.¹⁷ These factors can significantly complicate the learning process. In addition, the finite-horizon rollout in RL may introduce bias when estimating the value function.¹⁸ The aggregation of data into an experience replay memory may help to reduce non-stationarity, but it necessitates extensive computational resources and memory, and restricts the methods to off-policy algorithms.² Those challenges are further exacerbated when dealing with the unknown complex chemical reaction mechanisms, where the potential energy landscape (PEL) of chemical systems is characterized by strong nonconvexity, high noise, and high dimensionality. These characteristics present a significant hurdle for RL optimization. For example, RL exploration usually meets a great number of different reaction pathways, leading to a complex energy landscape with numerous local minima and similar energy barriers. These features can trap the policy exploration, preventing it from effectively escaping and causing the convergence to either be impossible or excessively slow.”

On page 8:

“In this reaction, we will explore the hydrogen migration mechanism in the Haber Bosch ammonia synthesis on the Fe(111) surface. The environment of this specific problem depends entirely on the coordination of the atoms within the system.”

(4) On p. 13, I think there was a misunderstanding regarding the "colormap". What I suggest is to replace the categorical color-coding of red -> green -> blue -> yellow with sequential colors. For example, the color coding could run from light blue (4) to dark blue (500) in Fig. 2 and 7. This would make it clearer that the authors are gradually increasing the number of concurrent instances.

Authors Reply:

We appreciate the suggestions regarding the colormap. However, we believe that using similar colors for different numbers of concurrent environment instances would result in poor contrast. In particular, some overlapping shadow regions (error bars) are hard to distinguish if we use a similar color coding (from light to dark). For reference, when this metric was first introduced, it was plotted in different colors for various concurrency levels

(Reference: Mnih, V. et al. Asynchronous methods for deep reinforcement learning. Proc. 528 Machine Learning Res. 48, 1928–1937 (2016)). Therefore, we have decided to retain our original color-coding in both Fig. 2 and Fig. 7.

(5) Related to a point in my previous report, it would be good to see a more helpful data and code availability statement. For example, the coordinate files for the structures are not provided, and these together with the code would allow a reader to reproduce the findings of the work (and ideally to help them to get started modeling other catalytic reactions!). It would also be good to provide the actual code used to train the models, for example in a Jupyter notebook, and the model parameters. The repository link currently refers to the general RL framework, but does not seem to include the particular code used in this work.

Authors Reply:

We thank the referee for the comment. We have updated the code availability and now share our code with the research community on GitHub. We added the sentence below in the code availability statement.

“The environment code is available at https://github.com/salesforce/warp-drive/tree/master/example_envs.”